# Haematopoietic stem cell gene therapy with IL-1Ra rescues cognitive loss in mucopolysaccharidosis IIIA

Helen Parker[1], Stuart M Ellison[1], Rebecca J Holley[1], Claire O'Leary[1], Aiyin Liao[1], Jalal Asadi[1], Emily Glover[1], Arunabha Ghosh[2], Simon Jones[2], Fiona L Wilkinson[3,4], David Brough[5] (iD), Emmanuel Pinteaux[5], Hervé Boutin[5,6] & Brian W Bigger[1,*] (iD)

## Abstract

Mucopolysaccharidosis IIIA is a neuronopathic lysosomal storage disease, characterised by heparan sulphate and other substrates accumulating in the brain. Patients develop behavioural disturbances and cognitive decline, a possible consequence of neuroinflammation and abnormal substrate accumulation. Interleukin (IL)-1β and interleukin-1 receptor antagonist (IL-1Ra) expression were significantly increased in both murine models and human MPSIII patients. We identified pathogenic mechanisms of inflammasome activation, including that disease-specific 2-O-sulphated heparan sulphate was essential for priming an IL-1β response via the Toll-like receptor 4 complex. However, mucopolysaccharidosis IIIA primary and secondary storage substrates, such as amyloid beta, were both required to activate the NLRP3 inflammasome and initiate IL-1β secretion. IL-1 blockade in mucopolysaccharidosis IIIA mice using IL-1 receptor type 1 knockout or haematopoietic stem cell gene therapy over-expressing IL-1Ra reduced gliosis and completely prevented behavioural phenotypes. In conclusion, we demonstrate that IL-1 drives neuroinflammation, behavioural abnormality and cognitive decline in mucopolysaccharidosis IIIA, highlighting haematopoietic stem cell gene therapy treatment with IL-1Ra as a potential neuronopathic lysosomal disease treatment.

**Keywords** cognitive decline; haematopoietic stem cell gene therapy; inflammasome; interleukin-1 receptor antagonist; mucopolysaccharidosis
**Subject Categories** Genetics, Gene Therapy & Genetic Disease; Musculoskeletal System; Stem Cells & Regenerative Medicine

## Introduction

Many lysosomal storage diseases exhibit pronounced substrate deposition, neuroinflammation and neuronal degeneration (Platt *et al*, 2012; Bosch & Kielian, 2015). Several studies have suggested that correction of neuroinflammation could ameliorate abnormal behaviours in neuronopathic lysosomal diseases including Krabbe disease (Reddy *et al*, 2011), Sandhoff disease (Lee *et al*, 2007) and mucopolysaccharidoses (MPS) (Sergijenko *et al*, 2013; Holley *et al*, 2018). However, the series of events from initial enzyme deficiency and substrate accumulation, through to clinical manifestation of disease pathologies, remain obscure in all of these diseases, making the choice of anti-inflammatory target unclear. Additionally, the role of lysosomal substrates in precipitating these responses also remains uncertain. In MPSIIIA, it is known that a lack of lysosomal enzyme *N*-sulfoglucosamine sulfohydrolase (SGSH), which degrades the complex sugar heparan sulphate (HS), results in a progressive clinical disease, characterised by hyperactivity and other behavioural abnormalities, developmental regression and cognitive decline (Wraith, 2002; Heron *et al*, 2011; Valstar *et al*, 2011; Wijburg *et al*, 2013; Shapiro *et al*, 2016). To date, there are no clinically approved treatments; however, several potential therapies are currently under clinical evaluation, including adeno-associated virus-driven SGSH, recombinant SGSH and lentiviral-mediated haematopoietic stem cell gene therapy.

In the MPSIIIA mouse model, the lack of SGSH results in intra- and extracellular accumulation of partially degraded and highly sulphated fragments of HS in all tissues, including the brain (Langford-Smith *et al*, 2012; Wilkinson *et al*, 2012; Sergijenko *et al*, 2013). HS accumulation is accompanied by widespread inflammation, with substantial glial activation. This is accompanied by accumulation of secondary storage substrates including GM2 and GM3 gangliosides, cholesterol, amyloid beta, α-synuclein and hyper-phosphorylated tau (Ginsberg *et al*, 1999; McGlynn *et al*, 2004; Hamano

1 Stem Cell and Neurotherapies, Division of Cell Matrix Biology & Regenerative Medicine, Faculty of Biology, Medicine and Health, Manchester Academic Health Science Centre, University of Manchester, Manchester, UK
2 Royal Manchester Children's Hospital, Manchester University Hospitals NHS Foundation Trust, Manchester, UK
3 Division of Biomedical Sciences, School of Healthcare Science, Manchester Metropolitan University, Manchester, UK
4 The Centre for Bioscience, Manchester Metropolitan University, Manchester, UK
5 Division of Neuroscience & Experimental Psychology, Faculty of Biology, Medicine and Health, Manchester Academic Health Science Centre, University of Manchester, Manchester, UK
6 Wolfson Molecular Imaging Centre, University of Manchester, Manchester, UK
*Corresponding author. Tel: +44 0161 306 0516; E-mail: brian.bigger@manchester.ac.uk

*et al*, 2008; Ohmi *et al*, 2009; Wilkinson *et al*, 2012; Beard *et al*, 2017). MPSIIIA mice exhibit hyperactivity and memory impairment (Gliddon & Hopwood, 2004; Langford-Smith *et al*, 2011; Sergijenko *et al*, 2013), similar to observations seen in patients. Here, we sought to determine whether lysosomal substrates played a role in CNS inflammation and the cognitive behaviour observed in MPSIIIA.

IL-1β is a key pro-inflammatory cytokine that is primarily produced in the periphery by immune cells but can also be synthesised by glia and neurons within the brain. The activation of TLR4 on immune cells by substrates, such as HS, leads to the expression of immature pro-IL-1β that is stored intracellularly (Takeuchi & Akira, 2010). However, a secondary inflammatory stimulus, which activates inflammasome complexes such as NLRP3, and the protease caspase-1, is required to drive the cleavage and release of mature, active IL-1β (Medzhitov, 2001; Takeda & Akira, 2004; Martinon *et al*, 2007; Lopez-Castejon & Brough, 2011). IL-1 signalling is mediated by the interleukin type 1 receptor (IL-1R1), and IL-1 binding to the IL-1R1 is blocked by the competitive IL-1 receptor antagonist (IL-1Ra), which inhibits all signal transduction (Weber *et al*, 2010). IL-1 signalling has been shown to play a key role in locomotor activity, explorative behaviour, anxiety and cognition (Kitazawa *et al*, 2011; Hein *et al*, 2012; Murray *et al*, 2013; Wohleb *et al*, 2014), and inhibition of IL-1 and inflammasome signalling has highlighted IL-1 as a key mediator in several neurological disorders, e.g. Alzheimer's disease, Parkinson's disease, stroke and multiple sclerosis (Halle *et al*, 2008; Jha *et al*, 2010; Heneka *et al*, 2013; Jesus & Goldbach-Mansky, 2014; Daniels *et al*, 2016; Sobowale *et al*, 2016; Dempsey *et al*, 2017).

MPSIIIB is a clinically indistinguishable HS storage disease to MPSIIIA, where pathogenic HS appears to have a role in neuroinflammation (Ausseil *et al*, 2008). MPSIIIB HS, acting through Toll-like receptor 4 (TLR4), induces tumour necrosis factor (TNF)-α and interleukin (IL)-1β expression in primary microglial cultures (Ausseil *et al*, 2008). MPSIIIB mice crossed with either TLR4 knockout mice or MyD88 knockout mice (the adaptor protein downstream of TLR4) show initial improvements in brain microgliosis, but these inflammatory defects reappear at 3 months of age, coinciding with the accumulation of secondary storage substrates. This suggests that HS is not solely responsible for inflammation and that the inflammasome may not be completely activated by HS alone. Conversely, MPSVII mice, which develop a more somatic disease, and as such, less HS and more dermatan sulphate (DS) storage, show decreased levels of TNF-α and IL-1β, with commensurate improvements in somatic disease when crossed with TLR4 knockout mice (Simonaro *et al*, 2010).

In terms of treating inflammatory manifestations of these diseases, broad spectrum anti-inflammatory therapies have shown some promise, with prednisolone showing correction of hyperactive behaviour in MPSIIIB mice despite only having reduced peripheral inflammation (DiRosario *et al*, 2009; Holley *et al*, 2018), and aspirin showing reduced brain cytokine profiles, including IL-1β in MPSIIIA (Arfi *et al*, 2011). Unravelling the inflammatory mechanisms responsible for cognitive decline and behavioural abnormalities in MPSIIIA and other lysosomal storage diseases is essential for the development of future therapies.

Here, we addressed the role of lysosomal substrates in IL-1 pathways and whether IL-1 had a direct role in cognitive decline

associated with MPSIIIA. We found that both the MPSIIIA murine model and patients with MPSIII had elevated levels of secreted IL-1β, suggesting inflammasome activation. We found that glial activation *in vitro* is primed by pathogenic 2-*O*-sulphated MPSIIIA glycosaminoglycans (GAGs) through TLR4 signalling. Secondary storage substrates drove IL-1β secretion through an NLRP3 inflammasome-dependent mechanism, but only when pre-primed with MPSIIIA GAG. We attenuated IL-1 signalling in MPSIIIA mice using either lentiviral-mediated haematopoietic stem cell gene therapy to over-express IL-1Ra or through the generation of IL-1R1-deficient MPSIIIA mice. Both approaches prevented working memory deficits and hyperactivity, and reduced brain glial activation, without any reduction in neuronal lysosomal storage. These data suggest that IL-1 is an important mediator in the MPSIIIA inflammatory cascade, and precipitates at least some of the abnormal behaviours observed. We highlight haematopoietic stem cell gene therapy using IL-1Ra as a potential anti-inflammatory therapy to treat cognitive decline in MPSIIIA and other neuronopathic lysosomal storage diseases.

## Results

### Interleukin-1-dependent inflammation is observed in MPSIIIA

An array of pathological and immunological markers was quantified in 9-month-old MPSIIIA mouse brain to establish an observational study of events that may be associated with IL-1-mediated neuroinflammation. Significant brain astrocyte (GFAP) and microglial (ILB4) reactivity were observed within cortical layers II-VI (Fig 1A) in MPSIIIA mice compared to wild-type (WT) mice, as previously described (Wilkinson *et al*, 2012). Quantitative PCR within whole brain extracts from WT and MPSIIIA mice revealed significant up-regulation of pro-inflammatory cytokines TNF-α (*Tnfa*; $P < 0.001$), IL-1β (*Il1b*; $P < 0.001$), IL-1α (*Il1a*; $P < 0.05$), IL-6 (*Il6*; $P < 0.001$) and IL-1Ra (*Il1rn*; $P < 0.001$) (Fig 1B).

We also found an analogous change in protein levels of inflammatory mediators in plasma and cerebrospinal fluid (CSF) from MPSIII patients. We found the inflammatory markers IL-1β and IL-1Ra to be significantly elevated in MPSIIIA, IIIB and IIIC patients when compared to healthy participants; this increase was observed in both plasma (Fig 1C and D) and cerebrospinal fluid for IL-1Ra (Fig 1E and F). There were no significant differences between subtypes.

### MPSIIIA GAG primes an intracellular IL-1β inflammatory response

Heparan sulphate is known to control inflammatory responses, including acting as a co-receptor for cytokines/chemokines, modulation of leucocyte–endothelium interactions and initiation of immune responses. To understand whether GAG may be responsible for IL-1-dependent inflammation in MPSIIIA, we delivered either MPSIIIA GAG, WT GAG (amounts administered based on quantity of HS detected; Fig EV1), bovine kidney HS, heparin, PBS or lipopolysaccharides (LPS), a known pro-inflammatory response initiator (Cunningham *et al*, 2005) into 4-month-old C57BL/6J mice by intravenous injection. One hour, 2 and 6 h after treatment, the expression of *Il1b* in the brain was measured by qPCR (Fig 2A).

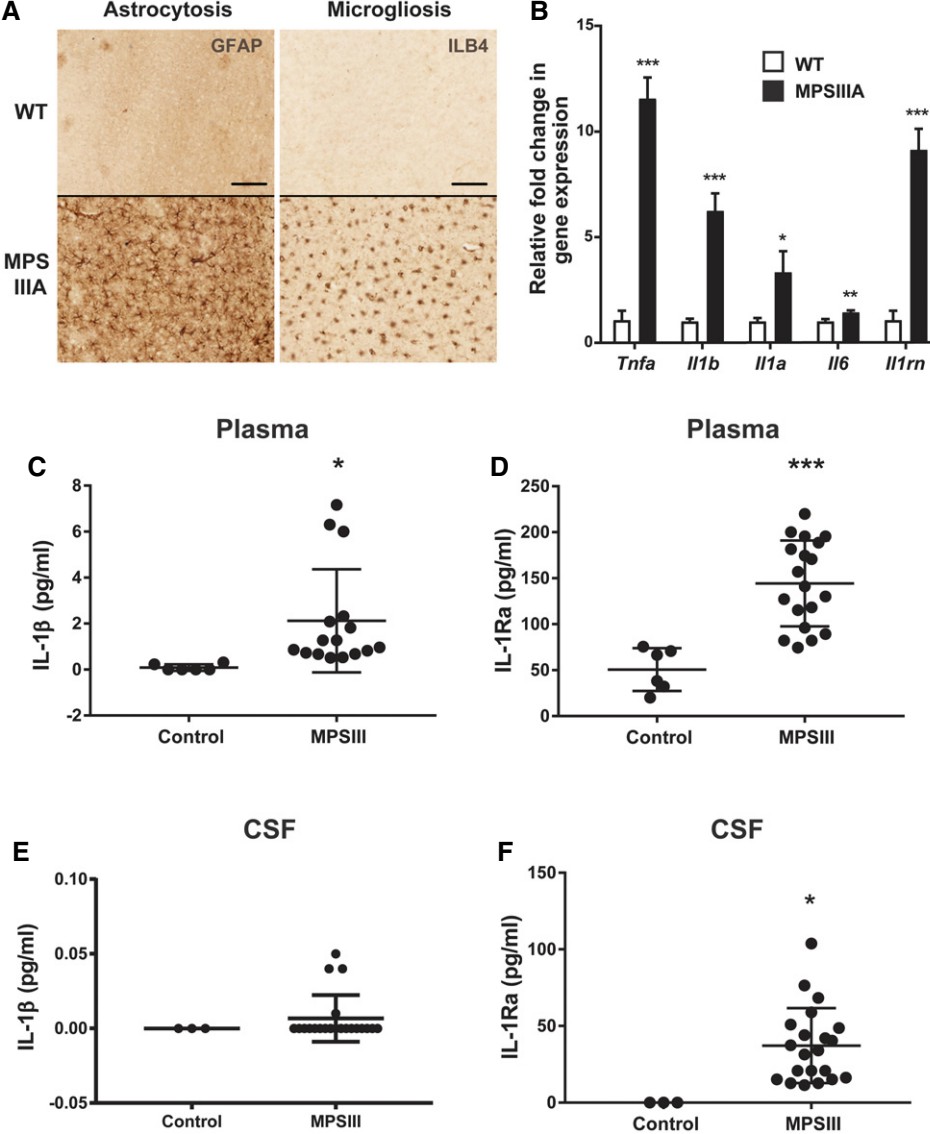

**Figure 1. Interleukin-1 and related cytokines are elevated in MPSIIIA mice and patients.**

A Representative sections of WT and MPSIIIA brains at 9 months of age stained for astrocytes (GFAP) or microglia (isolectin B4, ILB4). Images correspond to high power views covering cortical layers II-V. Scale bar, 100 μm (*n* = 4 mice per group).

B Quantitative PCR for various inflammatory cytokines in whole brains from 9-month-old WT and MPSIIIA mice (*n* = 6 mice per group). Data are expressed as mean ± STDEV, and data for each cytokine were tested by unpaired *t*-test; *P < 0.05, **P < 0.01, ***P < 0.001. Symbols above bars are versus WT. Exact *P*-values are indicated in Appendix Table S1.

C, D Quantification of IL-1β and IL-1Ra levels in plasma from human healthy control and MPSIII patients (*n* = 6 control participants and *n* = 21 MPSIII patients). Data are expressed as mean ± STDEV and were tested by unpaired *t*-test. (C) IL-1β, control versus MPSIII *P = 0.0407; (D) IL-1Ra, control versus MPSIII ***P = 0.0001.

E, F Quantification of IL-1β and IL-1Ra levels in cerebrospinal fluid (CSF) from control and MPSIII patients (*n* = 3 control participants and *n* = 21 MPSIII patients). Data are expressed as mean ± STDEV and were tested by unpaired *t*-test. (F) IL-1Ra, control versus MPSIII *P = 0.0167.

Mice treated with MPSIIIA GAG showed significant increases in *Il1b* gene expression 2 and 6 h post-treatment (*P* < 0.001), when compared to saline treatment (Fig 2B). *Il1b* gene expression in MPSIIIA GAG-treated mice significantly increased between 1 and 2 h, and declined by 6 h (Fig 2B). No response was achieved with WT GAG, control bovine kidney HS or heparin treatment whilst the positive control of LPS elicited significant responses at 1 h, declining steadily with time. Thus, MPSIIIA GAG specifically initiates an acute IL-1β inflammatory response *in vivo*.

IL-1β is initially produced in a pro-form, requiring inflammasome-mediated caspase 1 cleavage for activation and secretion. To explore whether GAGs alone stimulated the full IL-1β secretion response, equivalent amounts of MPSIIIA and WT GAGs were used to stimulate WT-mixed glial cultures *in vitro* (Fig 2C). Stimulation with MPSIIIA GAG significantly increased intracellular IL-1β production (*P* < 0.001) (Fig 2D), although no significant secretion of IL-1β into culture media was detected (Fig 2E) (below limit of detection) suggesting that GAGs alone prime IL-1β production.

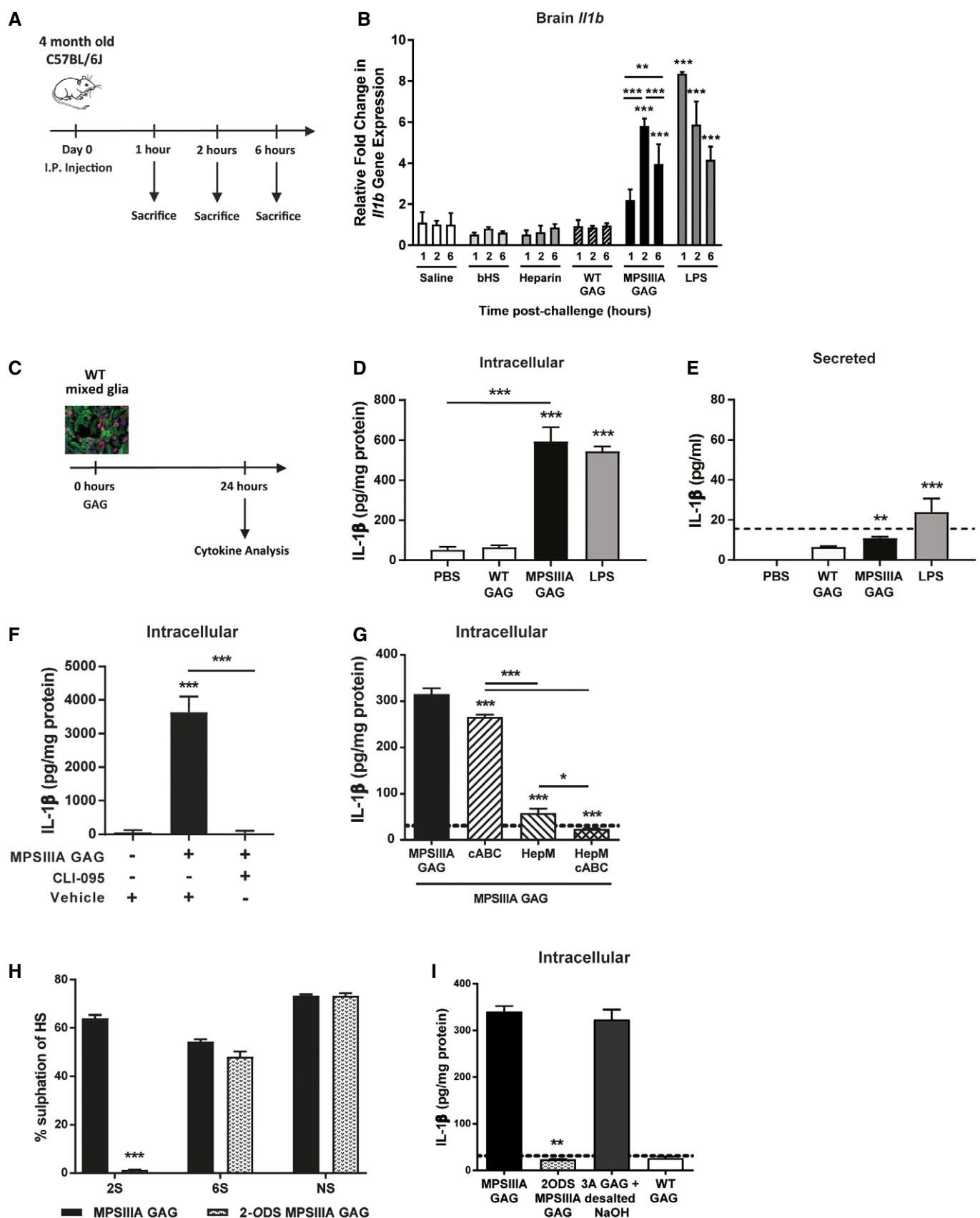

Figure 2.

**Figure 2. MPSIIIA 2-*O*-sulphated HS induces intracellular IL-1β expression *in vivo* and *in vitro*.**

A Mice were systemically challenged via intraperitoneal (I.P.) injection with PBS, bovine kidney HS (2.5 mg/kg), heparin (20 U/kg), WT GAG (2.5 mg/kg), MPSIIIA GAG (2.5 mg/kg) or LPS (250 μg/kg). Animals were sacrificed 1, 2 and 6 h after I.P. challenge.

B The expression of *Il1b* was measured in whole brain using quantitative PCR 1, 2 and 6 h after I.P. challenge (*n* = 6 mice per group). Data are expressed as mean ± STDEV and were tested by two-way ANOVA with Bonferroni's post-test; **P < 0.01, ***P < 0.001. Symbols above bars are versus PBS at the relevant time-point. Exact *P*-values are indicated in Appendix Table S2.

C *In vitro* modelling. WT or MPSIIIA GAG (0.8 μg/ml HS) was applied to WT-mixed glial cultures for 24 h before harvesting the media or cell extract and measuring levels of IL-1β.

D Quantification of intracellular production of IL-1β (*n* = 3 independent experiments each with 3 intra experimental replicates). Data are expressed as mean ± STDEV and were tested by one-way ANOVA with Tukey's post-test; ***P < 0.001. Symbols above bars are versus PBS. Exact *P*-values are indicated in Appendix Table S2.

E Quantification of IL-1β secreted into the media (*n* = 3 independent experiments each with three inter-experimental replicates). A horizontal dotted line represents the limit of detection of the ELISA. Data are expressed as mean ± STDEV and were tested by one-way ANOVA with Tukey's post-test; **P < 0.01, ***P < 0.001. Symbols above bars are versus PBS. Exact *P*-values are indicated in Appendix Table S2.

F WT-mixed glial cultures were co-treated with MPSIIIA GAG (4 μg/ml) and/or the TLR4 inhibitor CLI-095 (1 μg/ml) for 24 h. The intracellular production of IL-1β was measured (*n* = 3 independent experiments each with 3 intra experimental replicates). Data are expressed as mean ± STDEV and were tested by one-way ANOVA with Tukey's post-test; ***P < 0.001. Symbols above bars are versus MPSIIIA GAG + vehicle. Exact *P*-values are indicated in Appendix Table S2.

G Multi-heparinase (HepM) and/or chondroitinase ABC (cABC) digests were performed on MPSIIIA GAG, and the resulting oligosaccharides applied to a WT-mixed glial culture for 24 h alongside heparin, bovine HS (bHS), porcine DS (pDS) and WT GAG. The intracellular production of IL-1β was measured (*n* = 3 independent experiments each with three intra experimental replicates). A horizontal dotted line represents the limit of detection of the ELISA. Data are expressed as mean ± STDEV and were tested by one-way ANOVA with Tukey's post-test; *P < 0.05, ***P < 0.001. Symbols above bars are versus MPSIIIA GAG alone. Exact *P*-values are indicated in Appendix Table S2.

H MPSIIIA GAG was 2-*O*-desulphated (hashed bars) utilising 120 mM NaOH, and compositional HS disaccharide analysis performed via RP-HPLC against untreated MPSIIIA GAG (black bars) to validate complete 2-*O*-desulphation. Percentage contribution of each modification was determined (*n* = 3 GAG samples). Data are expressed as mean ± STDEV and were tested by two-way ANOVA with Bonferroni's post-test; 2S MPSIIIA GAG versus 2S 2-ODS MPSIIIA GAG ***P < 0.0001. *NS*, N-sulphated glucosamine; *2S*, 2-*O*-sulphate group; *6S*, 6-*O*-sulphate group.

I MPSIIIA GAG was 2-*O*-desulphated and the resulting oligosaccharides applied to a WT-mixed glial culture for 24 h. The intracellular production of IL-1β was measured (*n* = 3 independent experiments each with three intra experimental replicates). A horizontal dotted line represents the limit of detection of the ELISA. Data are expressed as mean ± STDEV and were tested by one-way ANOVA with Tukey's post-test; MPSIIIA GAG versus 2ODS MPSIIIA GAG **P < 0.0001. *2ODS*, 2-*O*-desulphated.

MPSIII pathogenic GAG has previously been shown to act via a TLR4-dependent response *in vitro* (Ausseil *et al*, 2008). When MPSIIIA GAG or LPS (positive control) was applied to WT-mixed glial cultures together with CLI-095 (a TLR4 intracellular domain inhibitor), complete abrogation of the inflammatory response was achieved, with significantly decreased intracellular production of IL-1β (*P* < 0.001), suggesting that GAGs do not act through any other innate immune receptor pathway (Fig 2F).

In order to determine whether the composition of MPSIIIA GAG was instrumental in eliciting the inflammatory response, MPSIIIA GAGs were digested with heparinases I, II and III (HepM) and/or chondroitinase ABC (cABC) to selectively remove HS and/or CS/DS, respectively and boiled to denature any co-purified GAG-binding proteins or lipids. Upon de-polymerisation of HS to constituent disaccharides, intracellular production of IL-1β was significantly reduced (*P* < 0.001; Fig 2G), to below the limit of detection. Degradation of CS/DS alone significantly reduced the intracellular production of IL-1β (*P* < 0.001; Fig 2G). Digestion of MPSIIIA GAG with both HepM and cABC resulted in a cumulative reduction in the production of IL-1β when compared to untreated MPSIIIA GAGs (*P* < 0.001) (Fig 2G). This suggests that the inflammatory response induced by MPSIIIA GAG can be largely attributed to HS but also partly to CS/DS.

2-*O*-sulphation of HS plays a role in protein-binding interactions and subsequent signalling; e.g. HS oligosaccharides containing 2-*O*-sulphated IdoA demonstrate a high-binding affinity for FGF2, IL-8, MCP-1 and other co-factors (Gallagher & Turnbull, 1992; Lortat-Jacob *et al*, 1995; Frevert *et al*, 2003; Simon Davis & Parish, 2013). To determine whether the 2-*O*-sulphated regions in MPSIIIA HS were involved in immune signalling, MPSIIIA GAG was selectively 2-*O*-desulphated whilst retaining other sulphate groups (Fig 2H). On application of 2-*O*-desulphated GAG to WT-mixed glial cultures, IL-

1β production was almost abrogated when compared to native MPSIIIA GAG (*P* < 0.001) (Fig 2I). Thus, 2-*O*-sulphation of MPSIIIA HS is essential for driving this inflammatory priming response.

These data confirm that priming of an intracellular pro-IL-1β (intracellular) inflammatory response is dependent on MPSIIIA GAG acting via TLR4. These data also suggest that it is specifically 2-*O*-sulphated HS, an HS composition that is enriched in MPSIIIA, that is critical for inflammatory activation of TLR4.

**MPSIIIA secondary storage substrates induce secretion of IL-1β via the NLRP3 inflammasome**

MPSIIIA GAG on its own does not initiate IL-1β secretion (Fig 2E), and yet in MPSIIIA mice and humans, we observe secreted IL-1β. This suggests that additional stimuli are required to initiate pro-IL-1β caspase-mediated cleavage and secretion. The activation of the NLRP3 inflammasome via damage-associated molecular patterns (DAMPs) results in NLRP3 oligomerisation into a caspase-1-activating scaffold, promoting the release of bioactive IL-1β (Chen & Nunez, 2010; Lopez-Castejon & Brough, 2011), and driving inflammation and pyroptosis (Lopez-Castejon & Brough, 2011).

To investigate the role of other secondary storage substrates accumulated in MPSIIIA and to understand whether the inflammasome was activated, we performed further pathological investigations in the mouse model of MPSIIIA. Immunostaining revealed significant GM2 ganglioside deposition within cortical layers II to V/VI of the cerebral cortex in MPSIIIA, which was negligible in WT (Fig 3A). Distribution of unesterified cholesterol, visualised using fluorescent filipin III, indicated extensive labelling of layers II and IV/V in the cortex (Fig 3A). Both of these are primary storage substrates in other neuronopathic lysosomal diseases. Amyloid-like protein aggregation, a hallmark of many late-onset neurodegenerative diseases, was

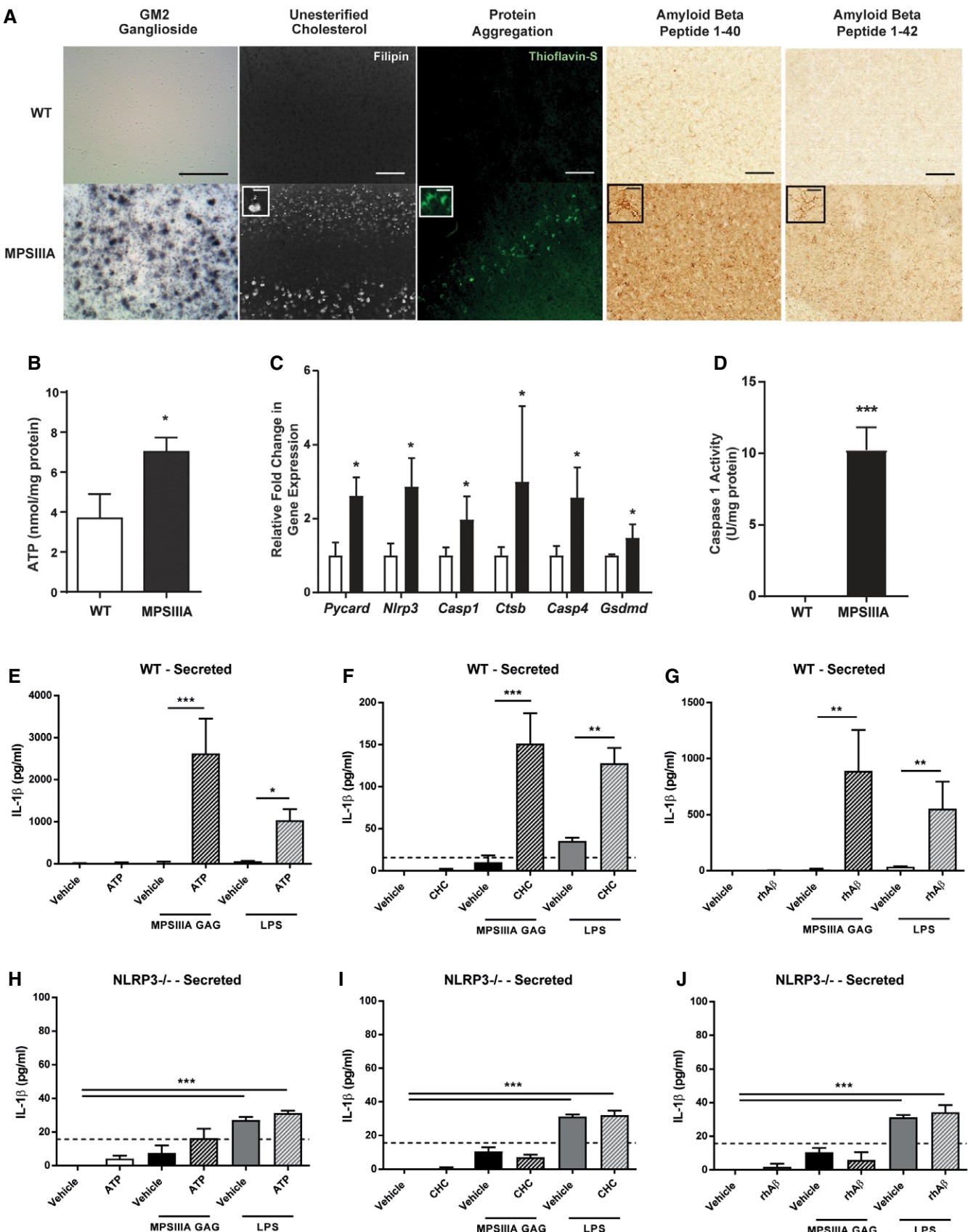

Figure 3.

**Figure 3.  Both HS and secondary stimuli are required to induce secretion of IL-1β via the NLRP3 inflammasome.**

A    Representative sections from 9-month-old WT and MPSIIIA brains showing secondary storage of GM2 ganglioside, cholesterol, protein aggregates and amyloid-beta peptide 1–40 and 1–42 within cortical layers II to V/VI, ×20, scale bar: 50 μm. Inserts, 100×, scale bar: 10 μm. ($n$ = 4 mice per group).

B    ATP levels in whole brain homogenate from 9-month-old WT and MPSIIIA mice ($n$ = 6 mice per group). Data are expressed as mean ± STDEV and were tested by unpaired $t$-test. WT versus MPSIIIA *$P$ < 0.02.

C    Brain transcription of mRNA species for various inflammasome components ($n$ = 6 mice per group). Data are expressed as mean ± STDEV, and data for each cytokine were tested by unpaired $t$-test; *$P$ < 0.05. Symbols above bars are versus WT. Exact $P$-values are indicated in Appendix Table S3.

D    Caspase 1 activity levels in whole brain homogenate from WT and MPSIIIA mice ($n$ = 6 mice per group). Data are expressed as mean ± STDEV and were tested by unpaired $t$-test. WT versus MPSIIIA ***$P$ < 0.0001.

E–J   WT (E–G)- and $Nlrp3^{-/-}$ (H–J)-mixed glia were primed with MPSIIIA GAG or LPS. ATP (5 mM), cholesterol crystals (1 mg/ml) or amyloid-β fibrils (Aβ$_{1-40}$ 4.3 μM and Aβ$_{1-42}$ 5 μM) were added 4 h post-priming stimulus, and the secretion of IL-1β into the media measured via ELISA after a further 20 h ($n$ = 3 independent experiments each with 3 intra experimental replicates). A horizontal dotted line represents the limit of detection of the ELISA. Data are expressed as mean ± STDEV and were tested by one-way ANOVA with Tukey's post-test; *$P$ < 0.05, **$P$ < 0.01, ***$P$ < 0.001. Exact $P$-values are indicated in Appendix Table S3.

Source data are available online for this figure.

detected via fluorescent thioflavin-S binding within layer V pyramidal neurons of the cortex (Fig 3A). More specifically, significant amyloid beta 1-40 and 1-42 reactivity were detected throughout the cortex, seemingly confined to neuronal cell bodies and glial cell membranes (Fig 3A). ATP is often released by activated immune and endothelial cells to potentiate inflammatory responses (Savage *et al*, 2012). An ATP assay revealed significant increases in ATP levels in MPSIIIA mice compared to WT ($P$ < 0.05; Fig 3B).

Significant increases in the expression of several subunits or activators of the NLRP3 inflammasome including Nlrp3, Pycard and Gsdmd were apparent in MPSIIIA brain tissue compared to WT (Fig 3C; $P$ < 0.05). This was confirmed with identification that downstream mediators of NLRP3 were also upregulated, including cathepsin B, caspase 4 and notably caspase 1 (Fig 3C; $P$ < 0.05), with functional activity of the latter, also significantly increased in MPSIIIA brain homogenate (Fig 3D; $P$ < 0.001). This suggests potential involvement of the inflammasome and activation of pyroptotic cell death, which requires caspase 1 activation.

To test the role of other lysosomal substrates in recapitulating IL-1-mediated inflammation, WT-mixed glia were primed with MPSIIIA GAG for 4 h, and secondarily stimulated with either ATP, cholesterol or amyloid-β for a further 20 h, with a readout of IL-1β secretion. ATP, cholesterol and amyloid-β were only able to drive significant IL-1β release ($P$ < 0.001; Fig 3E–G) following an MPSIIIA GAG priming stimulus, although each substrate on its own was able to prime intracellular IL-1β production in a similar way to GAG (Fig 3E–G). This suggests that although storage substrates can prime intracellular IL-1β-dependent inflammation, both primary and secondary storage substrates are required to activate IL-1β secretion. To determine the role of NLRP3 inflammasome activation in this process, these experiments were repeated in $Nlrp3^{-/-}$-mixed glial cultures that lack the critical component of the NLRP3 inflammasome. Upon MPSIIIA GAG priming and secondary stimulation with ATP, cholesterol or amyloid beta, no IL-1β secretion was apparent ($P$ < 0.001; Fig 3H–J), suggesting that the NLRP3 inflammasome is required for substrates to secrete IL-1β and drive a systemic inflammatory response.

Inflammasome activation occurs via multiple mechanisms, one of which involves lysosomal membrane destabilisation via substrate accumulation, lysosomal leakage of cathepsin B and subsequent NLRP3 inflammasome activation of caspase 1 to drive pro-IL1β cleavage and secretion (Rajamaki *et al*, 2010). WT-mixed glia were primed with MPSIIIA GAG and then treated with cholesterol 4 h post-priming in the presence or absence of caspase-1 inhibitor VX-

765, cathepsin B inhibitor CA-074Me, actin polymerisation/phagocytosis inhibitor cytochalasin D (required for cholesterol uptake) or an excess of KCl (required for NLRP3 activation). All four inhibitors significantly reduced IL-1β release ($P$ < 0.001; Fig EV2), demonstrating that caspase-1 and cathepsin B activity, alongside potassium efflux, contribute to cholesterol-dependent NLRP3 inflammasome-mediated IL-1β release.

## IL-1 is a major contributor to cognitive decline in a mouse model of MPSIIIA

To understand the role of IL-1 in chronic neuroinflammation, we attenuated IL-1 signalling in the mouse model of MPSIIIA using two complementary approaches. Firstly, we manufactured a lentiviral vector expressing the secreted isoform of human IL-1Ra under the control of the myeloid-specific CD11b promoter (Fig 4A). Successful expression of biologically active IL-1Ra was shown in transduced human RAW264.7 macrophages following treatment with LV.IL1RN or in non-transduced/control treated with IL-1β. The addition of IL-1β to LV.GFP-transduced macrophages induced significant secretion of TNF-α, indicating binding of IL-1β to IL-1R1 and downstream cytokine production, whereas no response was seen in LV.IL1RN-transduced cells, indicating that IL1Ra was biologically active and able to block IL-1β binding to its cognate receptor IL1R1 ($P$ < 0.001; Fig 4B).

Subsequently, CD45.1-MPSIIIA haematopoietic stem cells (HSCs) were transduced with LV.IL1RN at an MOI of 10 and transplanted into recipient CD45.2-MPSIIIA mice (Fig 4C). CFU assays were performed on transduced HSCs to assess HSC lineage development, toxicity, vector copy number and IL-1Ra expression. No lineage skewing or toxicity was apparent in CFUs (Fig EV3A); 3 viral copies were present in transduced cells (Fig 4D), with high IL-1Ra expression, both secreted (Fig 4E) and intracellular (Fig 4F). At 4 months of age (2 months post-transplant), 75–86% donor peripheral blood chimerism was achieved with a normal repertoire (Fig 4G and H) suggesting successful engraftment. High expression of hIL-1Ra was confirmed in the plasma (Fig 4I) in transplanted MPSIIIA mice. Significant viral integrations were detected in various post-mortem tissues (Fig 4J), with low levels of viral integrations evident in the brain. There was significant protein expression of hIL-1Ra in the brains of transplanted MPSIIIA mice when compared to non-transplanted controls (Fig 4K). To understand whether the increase in brain hIL-1Ra was peripheral or central, we performed immunohistochemistry and observed cell bodies expressing high levels of hIL-

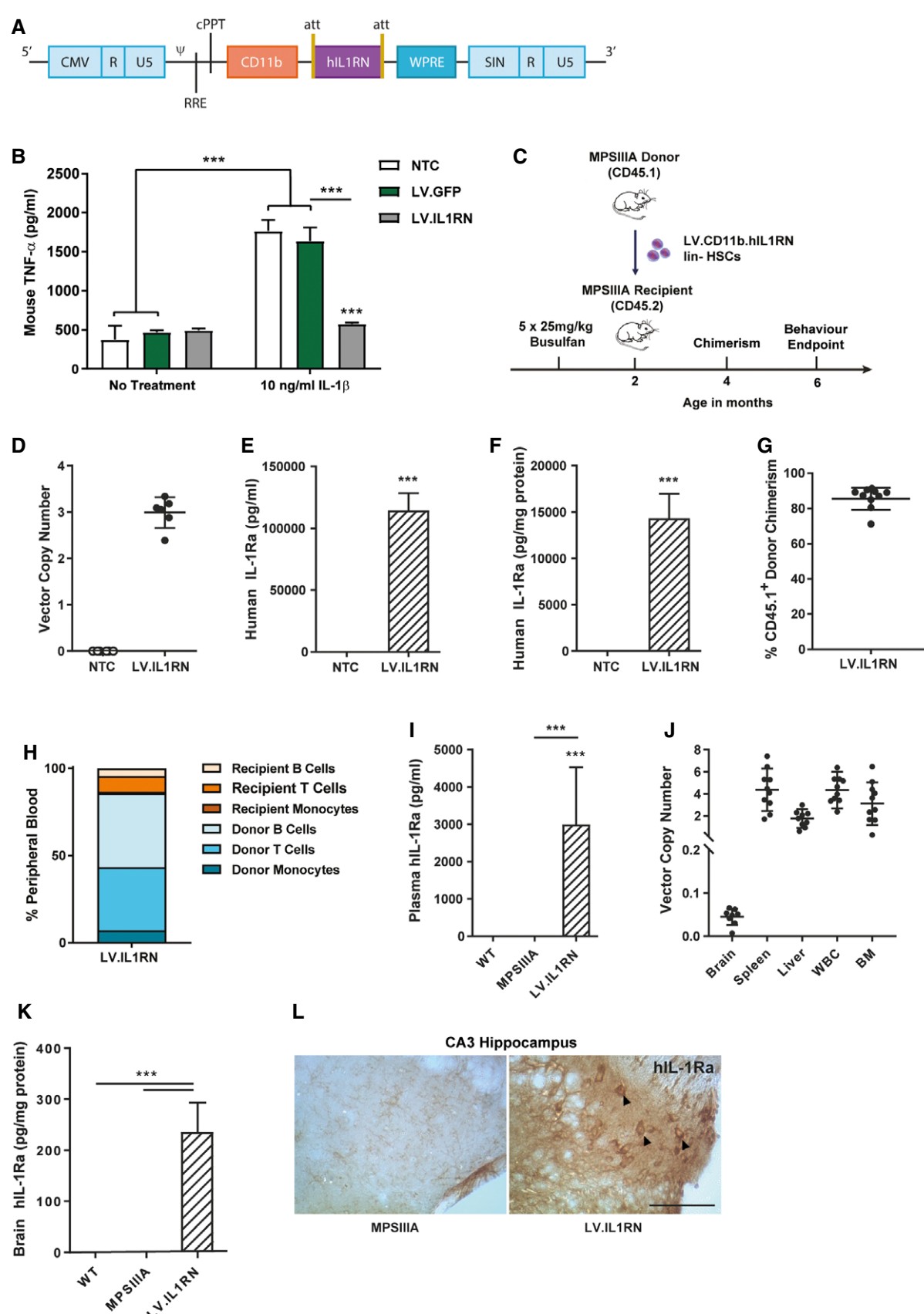

Figure 4.

**Figure 4. Evaluation of LV.IL1RN HSCT in MPSIIIA mice.**

A  Schematic of CD11b.IL1RN.WPRE lentivirus produced to drive codon optimised human IL1RN (secreted isoform).

B  The biological activity of LV-mediated IL-1Ra was assessed *in vitro*. RAW264.7 macrophages were transduced with LV.IL1RN or LV.GFP vectors at a MOI of 10. After 72 h, cells were stimulated with recombinant hIL-1β and the expression of TNF-α assessed after a further 48 h ($n$ = 3 independent experiments each with 3 intra experimental replicates). Data are expressed as mean ± STDEV and were tested by two-way ANOVA with Bonferroni's post-test; ***$P$ < 0.001. Symbols above bars are versus NTC within a treatment group. Exact $P$-values in Appendix Table S4.

C  Donor MPSIIIA bone marrow was lineage depleted and transduced with LV.IL1RN vector at a MOI of 10. 3 × $10^5$ transduced HSCs were transplanted into 6- to 8-week-old busulfan myeloablated recipient MPSIIIA mice. Chimerism was assessed 2 months post-transplant and behavioural assessment conducted 4 months post-transplant. Animals were sacrificed, and biochemical or histological analysis was carried out after behavioural assessment was finished.

D–F  A portion of transduced HSCs were plated into CFU assays, incubated for 12 days and CFU colonies analysed. Vector copy number (D) and protein production of secreted human IL-1Ra (E) and intracellular human IL-1Ra (F) were quantified ($n$ = 6 CFU assays). Data are expressed as mean ± STDEV and were tested by unpaired $t$-test. (E) WT versus MPSIIIA ***$P$ = 0.0001; (F) WT versus MPSIIIA ***$P$ = 0.0008.

G, H  Donor chimerism (G) based on peripheral blood composition (H) was determined by flow cytometry at 4 months of age (2 months post-transplant) ($n$ = 10 mice). Data are expressed as mean ± STDEV.

I  Human IL-1Ra expression was quantified in the plasma of mice at 6 months of age (4 months post-transplant) ($n$ = 10 mice per group). Data are expressed as mean ± STDEV and were tested by one-way ANOVA with Tukey's post-test; WT versus LV.IL1RN ***$P$ < 0.0001; MPSIIIA versus LV.IL1RN ***$P$ < 0.0001. Symbols above bars are versus WT.

J  Vector copy number was evaluated in brain, spleen, liver, white blood cells (WBC) and bone marrow (BM) of mice at 6 months of age (4 months post-transplant) ($n$ = 10 mice per group). Data are expressed as mean ± STDEV.

K  Quantification of human IL-1Ra levels in the brain of transplanted MPSIIIA mice ($n$ = 10 mice per group). Data are expressed as mean ± STDEV and were tested by one-way ANOVA with Tukey's post-test; WT versus LV.IL1RN ***$P$ < 0.0001; MPSIIIA versus LV.IL1RN ***$P$ < 0.0001. Symbols above bars are versus WT.

L  Representative sections from 6-month-old MPSIIIA and LV.IL1RN transplanted MPSIIIA mice. Sections show the CA3 region of the hippocampus. Cells expressing high levels of human IL-1Ra are indicated with black arrows. 40×, scale bar, 50 μm ($n$ = 4 mice per group).

1Ra throughout the brain and localised in the CA3 region of the hippocampus (Fig 4L).

In an independent approach, we inhibited IL-1 signalling by crossing MPSIIIA mice with mice lacking the main signalling IL-1 type 1 receptor (IL-1R1; IL-1R1$^{-/-}$ mice), which effectively abolishes all IL-1α and IL-1β downstream effects (Fig 5A). WT × IL-1R1$^{-/-}$ and MPSIIIA × IL-1R1$^{-/-}$ mice generated were viable and showed no overt differences in cognitive or neuropathological phenotype to MPSIIIA × IL-1R1$^{+/+}$ mice.

The Y-maze spontaneous alternation test indicates willingness to explore a new environment and evaluates spatial working memory (Fig 5B). At 6 months of age, MPSIIIA mice demonstrated a working memory deficit, as measured by the percentage of alternations in the Y-maze, compared to WT mice ($P$ < 0.05; Fig 5C). Chronic over-expression of LV-mediated hIL-1Ra or knockout of IL-1R1 prevented this working memory deficit in MPSIIIA mice ($P$ < 0.01 and $P$ < 0.05; Fig 5C). The total number of arm entries is an indicator of the exploratory behaviour of mice in a novel environment and acts as a proxy measure of anxiety. MPSIIIA mice exhibited significant increases in the number of arm entries ($P$ < 0.01; Fig EV3B), which was reversed by LV.IL1RN treatment ($P$ < 0.01). However, WT × IL-1R1$^{-/-}$ and MPSIIIA × IL-1R1$^{-/-}$ mice both showed increased arm entries when compared to WT ($P$ < 0.05).

It has previously been demonstrated that MPSIIIA mice are hyperactive and display reduced anxiety in the open field test (Langford-Smith *et al*, 2011) (Fig 5D and E). Chronic overexpression of LV-mediated hIL-1Ra or knockout of IL-1R1 in MPSIIIA mice prevented hyperactivity compared to MPSIIIA, with significant decreases in path length ($P$ < 0.001; Fig 5E).

The levels of astrogliosis and microglial reactivity were assessed by counting GFAP-positive astrocytes and isolectin B4-positive microglia in the brain primary somatosensory cortex and hippocampus at 6 months of age. Astrocyte and microglial activation were significantly increased in MPSIIIA compared to WT (Fig 6A–F). Overexpression of hIL-1Ra or global knockout of IL-1R1 in the MPSIIIA mouse reduced the number of activated microglia and astrocytes, and this decrease was more marked with regard to astrocytosis, indicating that interleukin-1 is a major driver of astrocytosis in MPSIIIA.

Lysosomal compartment size was assessed in neurons utilising the lysosomal marker LAMP2. No difference in lysosomal compartment size was evident between MPSIIIA control and MPSIIIA LV.IL1RN or MSPIIIA x IL-1R1$^{-/-}$ groups. Thus, correction of neuroinflammation is independent of storage substrate accumulation (Fig EV4A and B).

## Discussion

Here, we show that neuroinflammation and cognitive decline in MPSIIIA are driven by a two-step IL-1-dependent immune response mediated by the NLRP3 inflammasome. Both pathogenic primary HS substrates and secondary storage substrates are required to initiate the inflammasome and together mediate a robust pro-inflammatory IL-1 immune response. Blockade of IL-1 prevents the development of the neurocognitive behavioural phenotype seen in MPSIIIA mice and ameliorates neuroinflammation without affecting lysosomal storage.

Although neuroinflammation has long been identified in mouse models of all subtypes of MPSIII (McGlynn *et al*, 2004; Ausseil *et al*, 2008; Wilkinson *et al*, 2012; Martins *et al*, 2015), the role of cytokine elevation and in particular IL-1α, IL-1β and TNF-α elevation has been unclear, with conflicting hypotheses about which substrates may induce inflammation and through which pathways. The identification that pathologic HS could prime TLR4 in MPSIIIB and abrogate neuroinflammation was encouraging, but the later re-emergence of inflammation in that model suggested that HS activation of TLR4 was only one part of a greater immunological picture (Ausseil *et al*, 2008). A similar effect was observed in MPSVII mice, whereby skeletal pathology and serum TNF-α levels were reversed to WT levels by crossing MPSVII mice with TLR4 knockout mice. It is worth bearing in mind that bone and joint inflammation may be predominantly driven by TNF-α as opposed to IL-1. Moreover, soluble HS fragments, produced via cleavage with mammalian

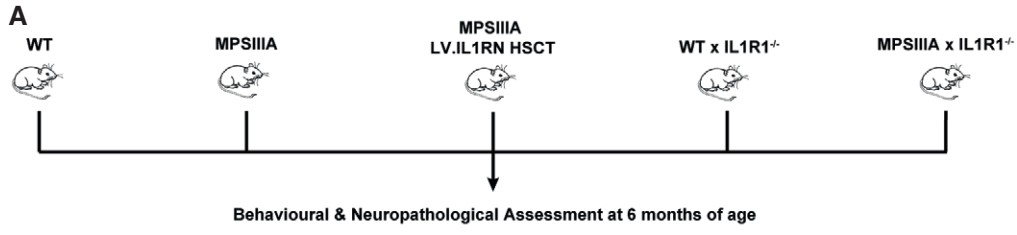

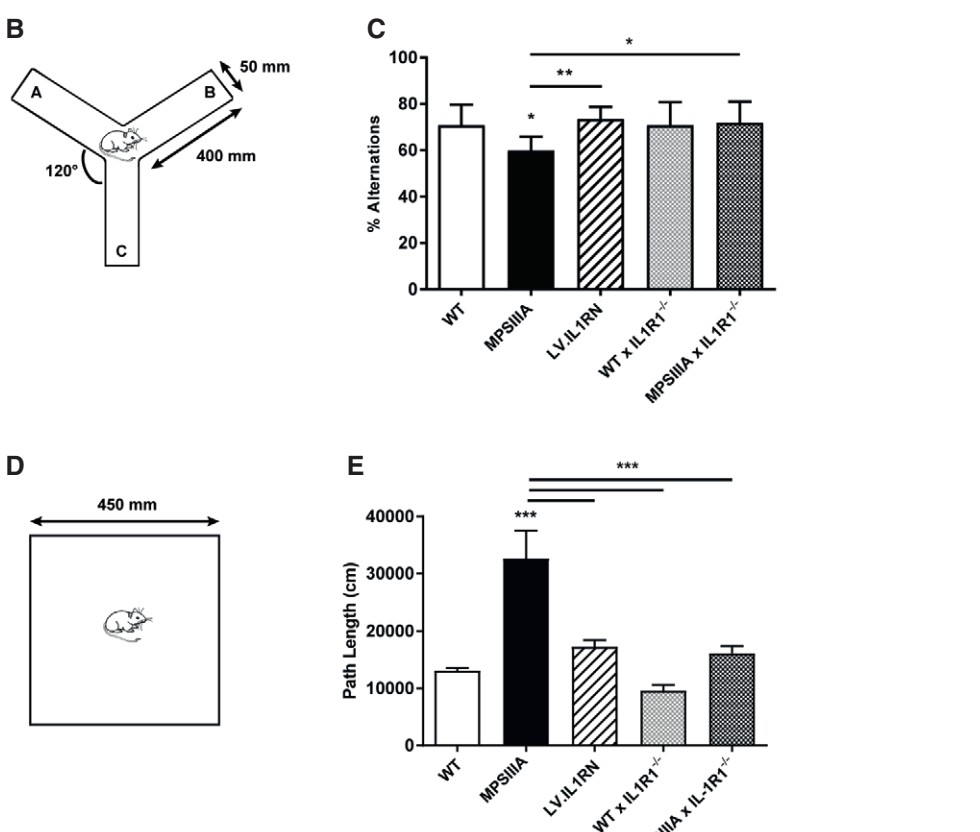

**Figure 5.  IL-1 signalling inhibition in MPSIIIA mice prevents memory deficits and hyperactivity.**

A     Behavioural assessments and histological analysis of control and treated mice from Fig 4 were carried out at 6 months of age in all groups, alongside WT × IL-1R1$^{-/-}$ and MPSIIIA × IL-1R1$^{-/-}$ mice.

B, C  Mice were placed in the centre of the Y-maze and allowed to explore freely for 10 min. The percentage of spontaneous alternations in the Y-maze were measured to assess working memory ($n = 10$ mice per group). Data are expressed as mean ± STDEV and were tested by one-way ANOVA with Tukey's post-test; $*P < 0.05$, $**P < 0.01$. Symbols above bars are versus WT. Exact $P$-values are indicated in Appendix Table S5.

D, E  A 1-h open field test was performed and path length measured as an indicator of hyperactivity ($n = 10$ mice per group). Data are expressed as mean ± STDEV and were tested by one-way ANOVA with Tukey's post-test; $***P < 0.001$. Symbols above bars are versus WT. Exact $P$-values are indicated in Appendix Table S5.

heparanase at regions of high sulphation, induced IL-1β, which was dependent on TLR4 signalling.

Although we also observed that MPSIIIA GAG primed an inflammatory response via TLR4 and that this induced intracellular IL-1β production, this was not sufficient to activate the NLRP3 inflammasome or stimulate IL-1β secretion. As greater understanding of the NLRP3 inflammasome has emerged (Medzhitov, 2001; Takeda & Akira, 2004; Martinon et al, 2007; Lopez-Castejon & Brough, 2011), it has become clear that a second activation signal is required to release active IL-1β from cells. Chronic secretion of IL-1β, as we observed in mouse models and for the first time, in the

plasma and CNS of MPSIII patients, may propagate a positive feedback loop of constitutive IL-1 activation. As patient cytokine levels fluctuate rapidly in response to multiple stimuli, including adventitious infection, either several measures of CSF or plasma from a patient or a relatively large cohort of 21 patients, as we have presented here, will be required to confirm consistent elevation.

We were also able to identify that 2-O-sulphation of HS, that is proportionally increased in MPSIIIA, was critical for TLR4-mediated signalling of HS and appears to provide the "pathogenic" factor in inflammation stimulation by HS. We and others have described increased 2-O-sulphation of HS in MPSI, MPSII, MPSIIIA, MPSIIIB

and MPSIIIC (Wilkinson *et al*, 2012; Gleitz *et al*, 2018; Tordo *et al*, 2018).

Our results demonstrate that both pathogenic 2-*O*-sulphated HS and other secondary storage substrates are required for the

initiation of IL-1β secretion via an NLRP3 inflammasome-dependent pathway *in vitro*. Endogenous MPSIIIA secondary storage substrates, ATP, cholesterol and amyloid beta (Aβ) were not only accumulated in the brains of MPSIIIA mice, but also activated the

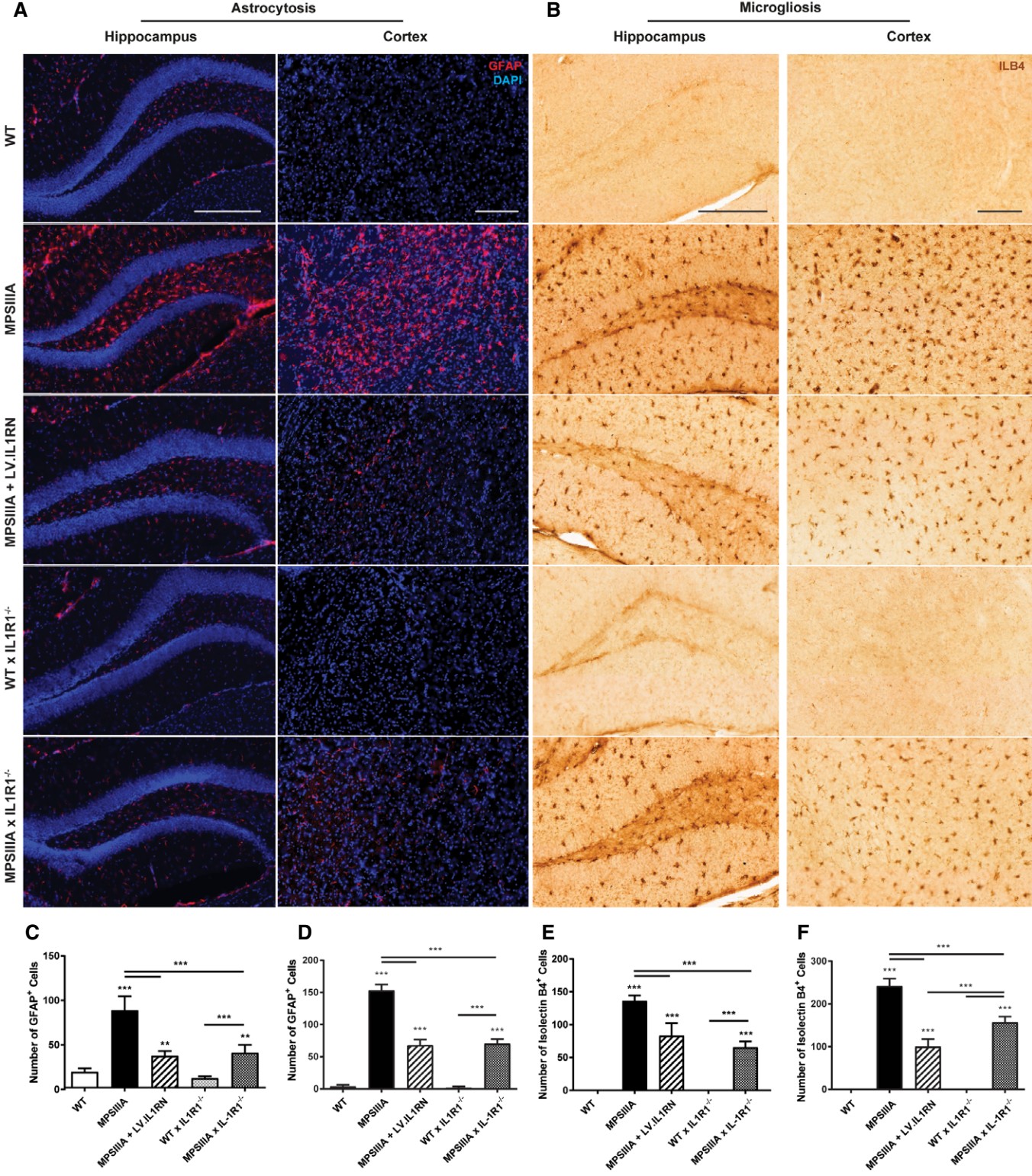

**Figure 6.**

**Figure 6. IL-1 signalling inhibition in MPSIIIA mice reduces glial activation.**

A Representative images of the hippocampus and cortex (layers II-V) from control and treated mice stained with anti-mouse glial fibrillary acidic protein (GFAP; red)/ DAPI (nuclear, blue) to identify activated astrocytes, 20×, scale bar: 50 μm.

B Representative images of the hippocampus and cortex (layers II-V) from control and treated mice stained with isolectin B4 (ILB4) to identify activated microglia, 20×, scale bar: 50μm

C, D The average number of GFAP-positive astrocyte cells over three fields of view per mouse of the hippocampus (C) and the cortex (D) was counted ($n$ = 4 mice per group, average of three fields of view per mouse). Data are expressed as mean ± STDEV and were tested by one-way ANOVA with Tukey's post-test; **$P$ < 0.01, ***$P$ < 0.001. Symbols above bars are versus WT. Exact $P$-values are indicated in Appendix Table S6.

E, F The average number of ILB4$^+$ microglial cells over three fields of view per mouse of the hippocampus (E) and the cortex (F) was counted ($n$ = 4 mice per group, average of 3 fields of view per mouse). Data are expressed as mean ± STDEV and were tested by one-way ANOVA with Tukey's post-test; ***$P$ < 0.001. Symbols above bars are versus WT. Exact $P$-values are indicated in Appendix Table S6.

Data information: Non-linear adjustments were made equally in fluorescent images to reduce background.

NLRP3 inflammasome as demonstrated by secretion of IL-1β *in vitro*.

Amyloid-beta deposits have already been evidenced in MPSIIIB cortices (Ohmi *et al*, 2011) and have been found co-localised to HS proteoglycans in MPSIIIA post-mortem brains (Ginsberg *et al*, 1999). Similarly, 2-*O*-sulphated HS iduronic acid residues bind fibrillary Aβ in Alzheimer's disease brains, whereby HS is able to modulate β-secretase activity and accelerate Aβ fibrillation (Castillo *et al*, 1999; Lindahl *et al*, 1999; Zhang *et al*, 2014; Liu *et al*, 2016). We observed secretion of IL-1β from MPSIIIA GAG-primed-mixed glial cultures after secondary treatment with fibrillary Aβ. Halle *et al* (2008) demonstrated swollen lysosomes with destabilised membranes upon phagocytosis of fibrillary Aβ by murine microglia (Halle *et al*, 2008). The activation of the NLRP3 inflammasome by lysosomal destabilisation increased release of IL-1β. However, the non-fibrillar form of Aβ did not induce secretion of IL-1β (Halle *et al*, 2008). Murphy *et al* also demonstrated that fibrillary Aβ treatment of primed rat glial cells increase cathepsin B activation, formation of the NLRP3 inflammasome, caspase 1 activation and IL-1β release (Murphy *et al*, 2014). It is unclear if fibrillary Aβ is a critical contributor to inflammasome activation in MPSIIIA, but as it is a major component of several neurodegenerative diseases, it should not be ruled out.

Accumulation of cholesterol occurs in various cells throughout the central nervous system and has been reported in a wide spectrum of LSDs, including not only MPS, but also Niemann-Pick C, GM1 and GM2 gangliosidosis, and α-mannosidosis (Walkley & Vanier, 2009), similar to the cholesterol accumulation demonstrated in MPSIIIA brains here. Our *in vitro* evidence has shown that IL-1β secretion can be triggered via inflammasome activation with cholesterol (Rajamaki *et al*, 2010). LPS primed monocytes and macrophages fed with cholesterol crystals have been shown to secrete IL-1β, in a similar fashion to our mixed glia (Rajamaki *et al*, 2010). Cathepsin B, caspase 1 and potassium efflux were also essential for the activation of NLRP3 inflammasome by cholesterol crystals as we observed. The mechanism by which cathepsin B activates the inflammasome remains unclear. It is likely that lysosomal leakage is via destabilisation of the lysosomal membrane, in response to lysosomal burden in MPSIIIA (Rajamaki *et al*, 2010). Increasing cholesterol synthesis in macrophages was also found to result in IL-1β secretion via an inflammasome-dependent pathway (Dang *et al*, 2017).

Whilst primary HS storage clearly initiates a priming of a danger-associated molecular pathogen (DAMP) type innate immune response in MPSIIIA, we cannot preclude the possibility that distention of lysosomes and resulting lysosomal membrane permeabilisation, releasing cathepsin B, may also be a secondary initiator of the NLRP3 inflammasome. It is no surprise therefore that cathepsin B inhibition has been shown to ameliorate cardiovascular pathology in MPSI mice (Gonzalez *et al*, 2018).

In summary, this work identifies that multiple primary and secondary stimuli can individually prime a DAMP response in MPSIIIA, resulting in intracellular IL-1 production, whilst two or more stimuli are required to activate the NLRP3 inflammasome in MPSIIIA, and propagate IL-1 secretion (Fig 7). Although we did not test many other combinations, the likelihood is that whether a substrate is primary or secondary is irrelevant for stimulation of NLRP3 inflammasome activation, suggesting that NLRP3 and IL-1 secretion will have an impact in most neuronopathic lysosomal diseases.

The central importance of an inflammatory axis initiated by the TLR4-NLRP3 pathway and IL-1β was emphasised by our finding that *in vitro* stimulation of NLRP3 null-mixed glia did not result in IL-1β secretion. In addition, glial activation is significantly reduced and behavioural abnormalities were prevented *in vivo* in MPSIIIA mice deficient in the IL-1R1 receptor for IL-1β or MPSIIIA mice over-expressing LV-mediated hIL-1Ra. We observed a pronounced reduction in astrogliosis upon inhibition of IL-1 signalling; astrocytes highly express IL-1R1, allowing them to respond to changes in the levels of central IL-1β (Rothwell & Luheshi, 2000; Basu *et al*, 2004; Srinivasan *et al*, 2004). In the diseased brain, increased production of pro-inflammatory cytokines and PGE$_2$ by activated astrocytes leads to reduced secretion of neurotrophic growth factors which can have a detrimental effect on long-term potentiation and neuronal functioning (Milligan & Watkins, 2009). At the same time, activated astrocytes can help to maintain damaged neurons; therefore, complete abrogation of astrogliosis could be detrimental. It is likely that astrocyte activation via chronic IL-1 overexpression affects neuronal functioning leading to cognitive dysfunction in MPSIIIA. Despite this, lysosomal storage dysfunction is still present; thus, astrocyte activation and microgliosis are likely required to help deal with damaged and distended neurons. The key is to reduce the chronic potentiation of this response by IL-1 (Fig 7). Microgliosis was less well corrected, but the primary driver here could be the presence of damaged cellular material from dysfunctional neurons full of lysosomal material.

Attenuation of IL-1 signalling provides an attractive target for therapeutic intervention to ameliorate the consequences of neuroinflammation, in particular preventing cognitive decline. Several studies indicate that the absence of IL-1 signalling is protective in central nervous system diseases characterised by neuroinflammation (Trittibach *et al*, 2008; van Strien *et al*, 2010; Murray *et al*, 2013; Wohleb *et al*, 2014; Clausen *et al*, 2016; Zhang *et al*, 2017). Application of IL-1Ra has demonstrated beneficial effects on the clinical

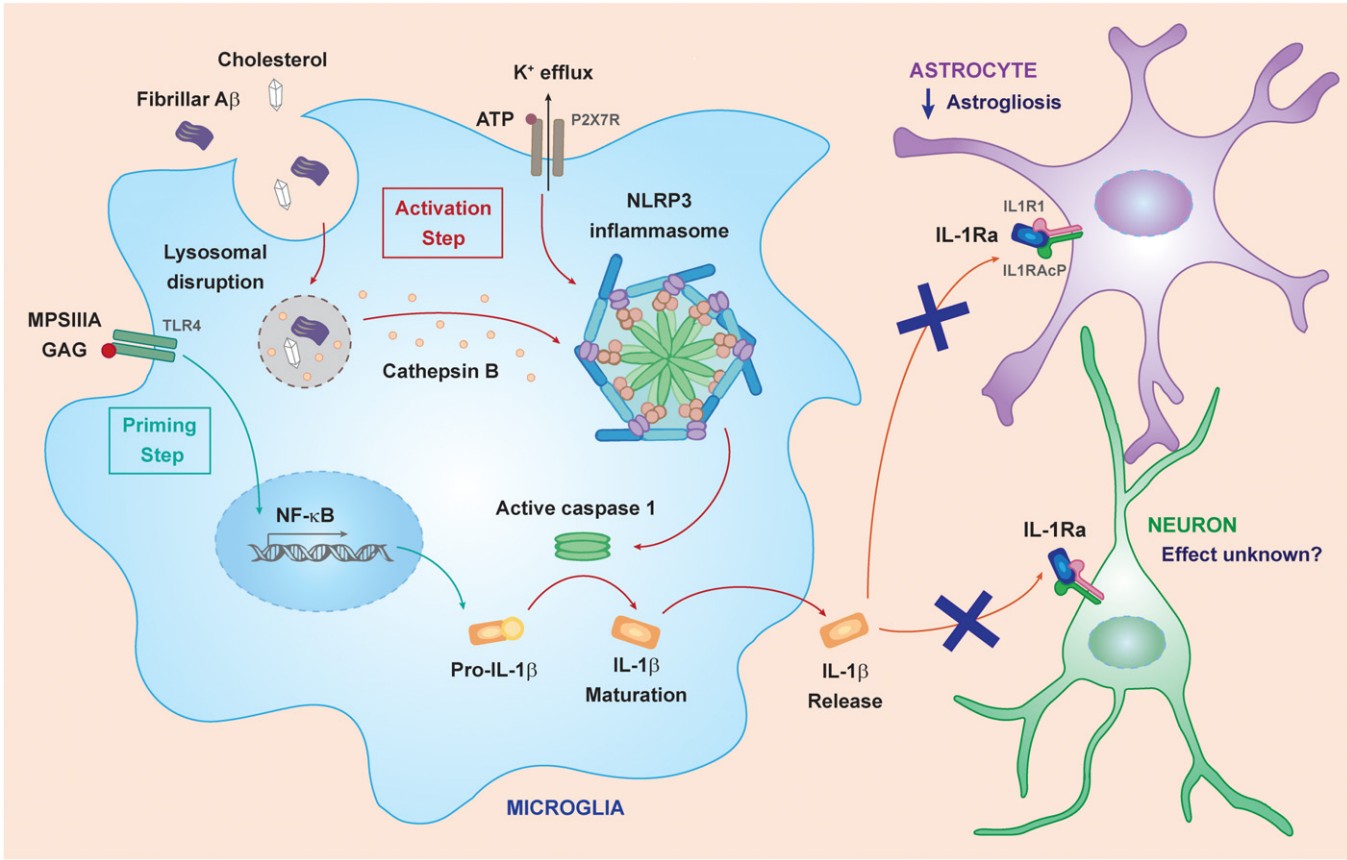

**Figure 7. Proposed role of IL-1 in neuroinflammation in MPSIIIA.**

MPSIIIA GAG primes a TLR4-mediated innate inflammatory response, resulting in production of intracellular pro-IL-1β (blue arrows). This step can potentially also be achieved by delivery of cholesterol crystals, amyloid-beta deposition or high levels of cellular ATP. Cholesterol, amyloid beta and ATP are each able to activate the formation of the NLRP3 inflammasome, usually via lysosomal destabilisation, inducing cleavage of pro-IL-1β by caspase 1 and secretion of IL-1β (red arrows), but only if cells are first primed. Extracellular IL-1β is able to bind IL-1R1 expressed on astrocytes and neurons, leading to astrocyte activation and changes in neuronal activity. Inhibition of IL-1 signalling reduces potentiation of microglial and astrocytic responses, presumably by reducing astrocyte activation, which leads to the prevention of behavioural and cognitive changes in MPSIIIA.

outcome of chronic-relapsing experimental autoimmune encephalomyelitis, traumatic brain injury and stroke (Emsley *et al*, 2005; van Strien *et al*, 2010; Smith *et al*, 2012; Helmy *et al*, 2014; Clausen *et al*, 2016; Zhang *et al*, 2017). Blockade of IL-1 signalling in the MPSIIIA mouse model led to the prevention of the working memory deficit, hyperactivity correction and reduced glial activation, particularly astrogliosis, supporting the role of IL-1 in the neuronopathic progression of MPSIIIA.

In conclusion, this study has delivered a comprehensive understanding of the pathophysiology of the neuronopathic lysosomal disease MPSIIIA. Given that both primary and secondary storage substrates were required to initiate NLRP3 inflammasome-mediated IL-1β secretion, it is likely that several other lysosomal storage diseases, accumulating similar substrates, will share a similar mechanism of neuroinflammatory activation. Questions remain as to the relationship between TNF-α-induced RIPK activation and necroptosis-mediated cell death described in Gaucher disease (Vitner *et al*, 2014) and Niemann-Pick C (Cougnoux *et al*, 2016), versus NLRP3-mediated IL-1 secretion and downstream effects described here in MPSIIIA, as well as inflammasome activation which has

also been reported in Gaucher macrophages (Aflaki *et al*, 2016). What is clear is that IL-1Ra-mediated blockade via haematopoietic stem cell gene therapy or via existing approved drugs could prove a therapeutic option for managing neuroinflammation and behavioural manifestations in MPSIIIA and potentially other early-onset lysosomal and CNS diseases, especially those lacking cross-correctable proteins.

# Materials and Methods

### Participants

Patient samples were gathered under the clinical trial GENISIS2013, EUDRACT number 2013-001479-18. Control samples were gathered under the ethics REC 08/H1010/63. Patients with MPSIII (subtypes A, B and C) ranging from 2 to 15 years of age had baseline cerebrospinal fluid (CSF)/plasma sampling performed before onset of the trial. IL-1β and IL-1Ra were evaluated in control and MPSIII human plasma and CSF using a commercially available Bio-Plex Pro™ Human Cytokine

27-Plex Assay (Bio-Rad Laboratories Ltd., Watford, UK) measured with a Bio-Plex® 200 system (Bio-Rad Laboratories Ltd.) powered by Luminex xMAP technology (Luminex, Austin, USA).

### Mice

Mice were housed in a 12/12-h light/dark cycle with food and water provided *ad libitum* in accordance with the Animal (Scientific Procedures) Act, 1986 (UK). All procedures were approved by the University of Manchester Ethical Review Process Committee. MPSIIIA mice on a mixed C57BL/6J background (B6.Cg-Sgsh^mps3a/6J) were maintained by heterozygote breeding, generating wild-type (WT) and MPSIIIA mice, genotyped as previously described (Bhaumik *et al*, 1999; Bhattacharyya *et al*, 2001). One-day-old WT pups (mixed gender) from this breeding were used to produce mixed glial cultures. Tissue from 9-month-old female WT and MPSIIIA mice was used for gene expression analysis, GAG isolation and substrate storage analysis. 6- to 8-week-old female MPSIIIA mice were used as recipients during HSCT. Due to conditioning with busulfan, recipient mice were kept in autoclaved cages throughout the transplantation protocol, and fed irradiated chow and acidified water. 6- to 8-week-old female WT and MPSIIIA mice were also used as controls for the HSCT *in vivo* study.

MPSIIIA mice backcrossed to PEP3 CD45.1 mice (B6.SJL-Ptprc^a Pepc^b/BoyJ; Jackson Laboratory) were used to distinguish donor cells from recipient as described (Langford-Smith *et al*, 2012). 6- to 12-week-old female MPSIIIAxPep3 offspring were used as donors for the HSCT *in vivo* study.

Six-month-old female MPSIIIAxIL1R1^−/− and WTxIL1R1^−/− mice were generated by crossing heterozygote MPSIIIA mice with IL-1R1^−/− mice (generated by Immunex, backcrossed onto C57BL/6 background by Dr Martin Nicklin (University of Sheffield) and generously provided by Dr Emmanuel Pinteaux). Offspring were inter-crossed to obtain third-generation WT and MPSIIIA mice deficient in IL-1R1 and genotyped as previously described (Parker *et al*, 2002). IL-1R1-deficient mice are kept in autoclaved cages and fed irradiated chow and acidified water due to an inherent level of immunodeficiency.

Nlrp3^−/− (Cias1^−/−) mice were generated by Dr Vishva Dixit (Genentech) (Mariathasan *et al*, 2006). We are grateful to Dr Dave Brough for providing Nlrp3^−/− 1-day-old pups (mixed gender) to prepare mixed glial cultures. 16-week-old female C57BL/6 mice were bought from Envigo, UK.

### Digestion, desulphation and biochemical analysis of heparan sulphate

GAGs were extracted and quantified from livers or brains from 9-month-old WT and MPSIIIA mice as described previously (Holley *et al*, 2018).

MPSIIIA GAGs isolated above were subject to heparinase I, II and III digestion and/or chondroitinase ABC digestions with two enzyme additions to ensure complete digestion. GAG samples were boiled for 20 min prior to use to deactivate enzymes and to denature co-purified GAG-binding proteins. For 2-*O*-desulphation, MPSIIIA GAG was incubated with 120 mM NaOH (pH < 13) for 20 min and then freeze-dried. Samples were desalted and subsequently freeze-dried. To confirm successful 2-*O*-desulphation, a combined heparinase I, II and III digest was performed, and the resulting disaccharides were 2-aminoacridone labelled and analysed by RP-HPLC.

### Isolation and treatment of primary mixed glial cultures

Primary mixed glial cultures were prepared from whole brains of 1-day-old mice (WT or *Nlrp3*^−/− mice) as described previously (Molina-Holgado *et al*, 2002), culturing for 14 days in DMEM/10% foetal bovine serum/2 mM glutamine/100 IU/ml penicillin/100 μg/ml streptomycin. Medium was changed every 72 h. At confluency, the cells were detached and cultured ($1.25 \times 10^5$ cells/cm$^2$) overnight before experiments. The cultures consisted of 65% astrocytes, 9.5% microglia and 25.5% other cells as determined by flow cytometry on a BD FACSCanto II flow cytometer (Das Sarma *et al*, 2009).

A priming stimulus of 0.8–4 μg/ml MPSIIIA GAG or WT GAG, 0.1–1 μg/ml lipopolyssacharides (LPS) (*Escherichia coli* 026:B6, Sigma) or phosphate-buffered saline (PBS) was added for a total of 24 h. Where indicated, 5 mM ATP (pH 7.4; Sigma), 1 mg/ml cholesterol crystals (CHC) (Rajamaki *et al*, 2010) or recombinant human amyloid-beta (Aβ) aggregates (a combination of 4.3 μM Aβ1-40 and 5 μM Aβ1-42; Invitrogen) were added to the incubation medium as specified in the Results sections. A cocktail of the recombinant human Aβ$_{1–40}$ (4.3 μM, aggregated for 24 h at 25°C; Invitrogen, Ireland) and Aβ$_{1–42}$ (5 μM, aggregated at 37°C for 48 h; Invitrogen, Ireland) was produced and fibrillisation assessed via an amyloid fibril dot blot (Rushworth *et al*, 2013). Endotoxin was not detected in GAGs, cholesterol crystals or Aβ aggregates by the Limulus Amoebocyte Lysate assay (Pierce™ LAL Chromogenic Endotoxin Quantitation Kit, Thermo Fisher Scientific). 1 μg/ml TLR4 inhibitor CLI-095 (InvivoGen), 50 μg/ml caspase-1 inhibitor VX-765 (InvivoGen), 2 μM cytochalasin D (Sigma), 10 mM cathepsin B inhibitor CA-074Me (Sigma) or 130 mM KCl was added to the incubation medium 0–4 h post-priming stimulus as described in results.

After *in vitro* stimulations, culture media were collected for analysis of the cytokine levels. Harvested cells were lysed with 50 mM Tris–HCl/150 mM NaCl/5 mM CaCl₂/0.02% NaN₃/1% Triton X-100, with Halt Protease Inhibitor Cocktail (pH 7.2–7.4) for 10 min on ice and centrifuged at 13,000 *g* for 15 min at 4°C to remove cellular debris.

### Cytokine analysis

Mouse IL-1β and human IL-1Ra were analysed from the culture media, the culture lysate and from mouse tissue samples using commercial enzyme-linked immunosorbent assays (ELISAs) (R&D Systems, Minneapolis, MN).

### Intraperitoneal challenge

Four-month-old female C57BL/6J mice (Envigo, UK; *n* = 4 per group) were injected intraperitoneally with 250 μg/kg of LPS, 2.5 mg/kg WT or MPSIIIA GAG, 2.5 mg/kg bovine kidney HS or 20 IU/kg heparin in 200 μl saline. Saline was administered as a control. Animals were sacrificed 1, 2 and 6 h post-injection. Mice were deeply anaesthetised and transcardially perfused with warmed phosphate-buffered saline (PBS). Brains were removed and snap-frozen for cytokine analysis by quantitative PCR. Plasma was collected for cytokine analysis by ELISA.

## Generation, titration and *in vitro* validation of LV.IL1RN

The codon optimised human IL1RN gene (secreted isoform; GenBank® accession number NM_173842.2) flanked with attB1/2 sites was inserted into a third-generation LV genome plasmid under the CD11b promoter (Sergijenko *et al*, 2013), to generate hCD11b.hIL1RN.WPRE (LV.IL1RN) using the Gateway® Recombination system (Life Technologies). GFP inserted in the same vector backbone, hCD11b.GFP.WPRE (LV.GFP), was used as a control *in vitro*. Lentivirus was produced and the titre determined as previously described (Langford-Smith *et al*, 2012; Sergijenko *et al*, 2013).

To assess the biological activity of lentiviral-mediated IL-1Ra protein delivery, RAW264.7 macrophages ($2.5 \times 10^4$ cells/cm², ATCC® TIB-71™) were transduced with LV.IL1RN or LV.GFP at a multiplicity of infection (MOI) of 10 as described (van Strien *et al*, 2010). The optimum dose of vector was determined based on previous findings which stipulate that a 100- to 1,000-fold molar excess of IL-1Ra protein is required to effectively block the biological effect of IL-1 (Arend *et al*, 1990; Sobowale *et al*, 2016). Media were changed 4 h post-transduction. Transduced cells were stimulated with 10 ng/ml recombinant human IL-1β (R&D systems) 72 h post-transduction and media removed 48 h following stimulation. Levels of TNF-α protein were assessed by ELISA (van Strien *et al*, 2010).

## Mouse HSC transduction and transplantation

Bone marrow was isolated from donor PEP3 CD45.1xMPSIIIA mice (6- to 12-week-old females), lineage depleted, transduced with an MOI of 10 and injected into 6- to 8-week-old busulfan-treated mice as described previously (Sergijenko *et al*, 2013). Briefly, recipient MPSIIIA mice (6- to 8-week-old females) were treated with intraperitoneal 125 mg/kg busulfan (Busilvex; Pierre Fabre, Boulogne, France) over 5 days (Sergijenko *et al*, 2013), prior to tail vein injection of $3 \times 10^5$ LV.IL1RN transduced haematopoietic stem cells (HSCs). Secretion of human IL-1Ra (hIL-1Ra) from transduced HSCs was assessed with ELISA as above, following plating of cells in colony-forming unit assays (Langford-Smith *et al*, 2012).

HSC engraftment (donor chimerism) was assessed at 2 months post-transplant in peripheral blood following staining with anti-mouse CD45.1-PE (donor) and CD45.2-FITC (recipient) (BD Pharmingen) antibodies and analysed on a BD FACSCanto II flow cytometer. White blood cell vector copy number and human IL-1Ra expression in plasma via ELISA were assessed 2 months post-transplant. Tissue vector copy number was assessed via qPCR as previously described (Langford-Smith *et al*, 2012; Sergijenko *et al*, 2013; Holley *et al*, 2018).

## Behavioural assessment

Behavioural assessment was performed at 4 months post-transplant. One and a half hours into the 12-h light phase, mice were placed into the centre of an open field arena as previously described (Langford-Smith *et al*, 2011), recording hyperactive behaviour for 60 min and analysing using EthoVision XT software (Noldus Information Technology). Spontaneous alternation was assessed during one continuous 10-min session in a Y-maze consisting of three identical arms as previously described (O'Leary *et al*, 2014; Gleitz *et al*, 2017).

## Tissue harvesting

At 6 months of age, mice were deeply anaesthetised and transcardially perfused with warmed PBS. Citrated blood was collected via cardiac puncture, centrifuged at $300 \times g$ for 10 min and plasma stored at $-80°C$ before the perfusion. Following perfusion, brains were removed and either snap-frozen and stored at $-80°C$ for biochemical analysis or fixed in 4% paraformaldehyde in PBS for 24 h at 4°C, followed by cryopreservation in 30% sucrose/2 mM $MgCl_2$/PBS for 48 h at 4°C before storing at $-80°C$ for histological analysis (Wilkinson *et al*, 2012).

## Immunohistochemistry

Brain sections (30 μm) were cut using a freezing sliding microtome (Hyrax S30, Zeiss, Germany). Free-floating immunohistochemistry was performed on sections taken from Bregma 0.98, 0.14, $-0.46$, $-1.06$, $-1.70$ and $-2.40$ according to the mouse brain atlas (McGlynn *et al*, 2004; Canal *et al*, 2010; Malinowska *et al*, 2010; Langford-Smith *et al*, 2012). Antigen retrieval was performed prior to amyloid-beta staining with 70% formic acid. Primary antibodies included rabbit anti-glial fibrillary acidic protein (GFAP 1:1500; DakoCytomation, Ely, UK), mouse anti-GM2 (1:500; a gift from Dr Konstantin Dobrenis and Prof Walkley), anti-β-amyloid 1–40 antibody clone 11A50-B10 1:250 (BioLegend), anti-β-amyloid 1–42 antibody clone 12F4 (1:1,000; BioLegend), goat anti-human IL-1Ra (5 μg/ml; R&D Systems), rat anti-lysosome-associated membrane protein 2 (LAMP2; 1:500 Abcam), rabbit anti-neuronal nuclei (NeuN; 1:200 Abcam) and peroxidase-conjugated isolectin B4 from *Bandeiraea simplicifolia* (ILB4; Sigma, Poole, UK). Secondary antibodies used included biotinylated goat anti-mouse IgM, biotinylated goat anti-rabbit IgG and biotinylated goat anti-mouse IgG (Vector laboratories), as well as goat anti-rat Alexa Fluor 488 and goat anti-rabbit Alexa Fluor 546 (Thermo Fisher Scientific) (mainly at 1:1,000). The VECTASTAIN ABC system and 3,3′-diaminobenzidine substrate (Vector Laboratories) were used to detect biotinylated antibody binding. For GM2 staining, nickel ions were included in the 3,3′-diaminobenzidine substrate.

To visualise unesterified cholesterol, sections were incubated with 50 μg/ml filipin III (from *Streptomyces filipinensis*; Sigma) at room temperature for 1 h (McGlynn *et al*, 2004; Phillips *et al*, 2008). Protein aggregates including amyloid-beta plaques were visualised via thioflavin-S histochemistry as previously described (Smith *et al*, 2014).

3,3′-Diaminobenzidine-stained sections were mounted onto positively charged slides (Fisher Scientific, Loughborough, UK) and mounted in DPX medium (Fisher Scientific). Fluorescently stained sections were treated with 4′,6-diamidino-2-phenylindole (DAPI) to label nuclei and mounted onto positively charged slides in ProLong Gold Antifade Mountant (Thermo Fisher).

Images were acquired on a 3D-Histech Pannoramic-250 microscope slide-scanner with 20×/0.30 Plan Achromat objective (Zeiss) or FITC, TRITC and DAPI filter sets, processed using Case Viewer (3D-Histech) and analysed using ImageJ (NIH) software (Wilkinson *et al*, 2012; Holley *et al*, 2018).

**The paper explained**

**Problem**

There is currently no cure for patients with the neuronopathic disease mucopolysaccharidosis IIIA. Anti-inflammatory therapies have shown some promise in alleviating clinical symptoms; however, long-term use of many anti-inflammatories and steroids is contraindicated.

**Results**

We demonstrate that interleukin-1 has a central role in the cognitive and behavioural phenotype seen in MPSIIIA. IL-1 signalling inhibition via knock out of IL-1 receptor 1 or IL-1 receptor blockade with its cognate inhibitor IL-1Ra, via stem cell gene therapy, alleviates neuroinflammation and cognitive decline in MPSIIIA mice.

**Impact**

IL-1 is a critical mediator of the MPSIIIA inflammatory cascade. Interleukin-1 inhibition is a potential anti-inflammatory therapy to treat cognitive decline and immunopathology in MPSIIIA.

## Quantitative RT–PCR

Total RNA was extracted from 9-month-old WT or MPSIIIA brains utilising the TRIzol® reagent RNA isolation procedure (Life Technologies). Total RNA was reverse transcribed into cDNA using the High Capacity cDNA Reverse Transcription Kit (Applied Biosystems) and RNase inhibitor. Quantitative PCRs using 20 ng cDNA, 2× TaqMan Gene Expression Master Mix and gene-specific 20× TaqMan Gene Expression Assays (Table EV1) were performed, according to the manufacturer's instructions (Applied Biosystems). Fold changes in gene expression were calculated as the ratio of molecules of the target gene against the housekeeping gene glyceraldehyde 3-phosphate dehydrogenase (GAPDH), via $\Delta\Delta C_T$ analysis.

## Caspase 1 activity assay

Caspase 1 activity was measured using 100 μg total brain protein with the Caspase 1 Fluorometric Assay Kit (ab39412) as per the manufacturer's protocol (Abcam, UK), using recombinant mouse caspase 1 (0.02-2U; Abcam) standards.

## ATP assay

The amount of ATP was measured using the ATP Assay Kit (ab83355) as per the manufacturer's protocol (Abcam) and included sample PCA/KOH deproteinisation (Hering *et al*, 2015).

## Experimental design and statistical analysis

Behavioural assessments were carried out on five groups of mice ($n = 10$ mice per group). Histology and biochemistry were carried out on $n = 4$–10. $N$ numbers were based on previous power calculations (Langford-Smith *et al*, 2012; Sergijenko *et al*, 2013; Gleitz *et al*, 2018; Holley *et al*, 2018). Criteria for sample exclusion were pre-established with regard to animals receiving HSCT. Any animal where chimerism was below 50% was excluded from the analysis, as this was deemed a failed transplant. No animals were excluded from the analysis. Animals were not caged based on genotype; WT, MPSIIIA littermates were mixed and WTxIL1R1$^{-/-}$ and

MPSIIIAxIL1R1$^{-/-}$ littermates were mixed. Animals and samples were assigned identification numbers to minimise subjective bias. Biochemical and microscopic analyses were performed blinded using animal identification numbers. Treatment of animal groups was impossible to completely blind due to the phenotype of the animal model and the treatments administered. Behavioural analyses were recorded and analysed at a later time-point from video in a blinded fashion once all tests had been performed. Treatment groups were filled equally to negate bias. All animal numbers were randomised prior to the analysis and subsequently allocated to correct groups.

All values are expressed as means ± standard deviation (STDEV). Student's *t*-test, one-way ANOVAs (with Tukey's post hoc multiple comparison analysis) or two-way ANOVAs (with Bonferroni's post hoc multiple comparison analysis), as appropriate, were applied for comparisons of differences between experimental groups. Normality was tested via a D'Agostino-Pearson normality test (omnibus K2). Where data were skewed, raw values were log transformed and statistical analysis carried out on the transformed data. Statistical analysis was performed using GraphPad Prism 7 and IBM SPSS Statistics 22 software. Values of $P < 0.05$ were considered significant.

**Expanded View** for this article is available online.

## Acknowledgements

The authors thank the staff of the Manchester BSF for assistance. The Bioimaging Facility microscopes used in this study were purchased with grants from BBSRC, Wellcome and the University of Manchester Strategic Fund. Special thanks to Roger Meadows for his help with microscopy. We gratefully acknowledge Shire Plc for providing the recombinant human SGSH and thank Dr. Vishva M. Dixit at Genentech Inc for providing *Nlrp3*$^{-/-}$ mice. HP was supported by a PhD studentship awarded by the Neuroscience Research Institute, University of Manchester.

## Author contributions

HP performed experiments, analysed data and wrote the manuscript. All authors provided critical analysis of the data and contributed to the manuscript. HP, HB and BWB designed the study. SME contributed to lentiviral cloning and production of lentivirus. RJH performed RP-HPLC. JA contributed to lentiviral cloning. EG performed the *in vitro* LV.IL1RN activity assay. COL and AL contributed to the *in vivo* studies. FLW, DB and EP contributed to the study design. AG and SJ were the clinicians involved in patient sample acquisition. BB obtained funding.

## Conflict of interest

BB has shares and licensed programmes in enzyme replacement stem cell gene therapy for MPSIIIA and MPSIIIB to Orchard Therapeutics Ltd. BB has shares and licensed programme in enzyme replacement gene therapy for MPSIIIC to Phoenix Nest Inc. Neither conflict of interest competes with the content of this paper, which deals with pathophysiology- and inflammatory-based treatment of MPSIIIA and other LSDs.

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
