## [Review Process File · EMBO Molecular Medicine]

Haematopoietic stem cell gene therapy with IL-1Ra rescues cognitive loss in mucopolysaccharidosis IIIA

Helen Parker, Stuart M Ellison, Rebecca J Holley, Claire O'Leary, Aiyin Liao, Jalal Asadi, Emily Glover, Arunabha Ghosh, Simon Jones, Fiona L Wilkinson, David Brough, Emmanuel Pinteaux, Herve Boutin & Brian W Bigger

Review timeline:

Submission date:	24 July 2019
Editorial Decision:	9 September 2019
Revision received:	12 December 2019
Editorial Decision:	20 December 2019
Revision received:	15 January 2020
Accepted:	17 January 2020

Editor: Céline Carret

Transaction Report:

1st Editorial Decision

9 September 2019

Thank you for the submission of your manuscript to EMBO Molecular Medicine. We have now heard back from the two referees whom we asked to evaluate your manuscript.

You will see from the set of comments pasted below that the two referees are overall supportive of publication. While referee 2 recommends rewriting and better structuring the paper, referee 1 is a little more critical and mentions a few technical issues, the need for clarifications and some supporting experiments that should be easily addressable. This referee also would like to see some data aimed at identifying the causal factor responsible for the observed effects. Should you have such data at hand, I would strongly encourage you to add them, however those will not be mandatory for acceptance.

We would therefore welcome the submission of a revised version within three months for further consideration and would like to encourage you to address all the criticisms raised as suggested to improve conclusiveness and clarity. Please note that EMBO Molecular Medicine strongly supports a single round of revision and that, as acceptance or rejection of the manuscript will depend on another round of review, your responses should be as complete as possible.

I look forward to receiving your revised manuscript.

***** Reviewer's comments *****

Referee #1 (Comments on Novelty/Model System for Author):

the experiments are well performed, results are highly relevant for understanding neuropathology in MPS. The mouse model recapitulates the phenotype seen in patients.

Referee #1 (Remarks for Author):

In the manuscript by Parker et al., the involvement of interleukin-1beta and interleukin-1 receptor antagonist in neuroinflammation induced by MPSIIIA is described. Using lentiviral gene therapy, the authors describe an approach for reduction of neuroinflammation in a mouse model. In general, the findings are interesting and provide novel insight in the pathogenesis of MPSIIIA.

Comments:

- 1) In the in vitro experiments using GAGs from various sources, the authors show that MPSIII GAGs specifically induced Il1b expression, whereas wt GAGs or heparin sulfate does not. While interesting, the authors do not identify the causal factor. What is in the MPSIII GAGs preparation that causes this effect?
- 2) Fig 2B: what was the effect of GAG treatment at later time points?
- 3) In Fig. 2E, the authors state that no significant secretion of Il-1b into culture media was detected. However, the figure does show significant secretion.
- 4) In Fig 3C, how was expression normalized? please show not-normalized expression levels of the genes examined as well as of the gene used for normalization
- 5) In Fig 4J, the VCN in brain is very low. Please comment.
- 6) please provide a more extensive analysis of Il1b expression showing where it is expressed in the brain
- 7) page 15, 3rd paragraph: the authors state that MPSIIIA x IL-1R1^{-/-} mice showed no overt differences in cognitive or neuropathological phenotype to MPSIIA x IL-1R1^{+/+} mice. Please specify the data. It sounds strange in the light of the results of Fig. 6
- 8) Fig. 6: the reduction of astrocyte or microglial activation by overexpression of hIL-1Ra or IL-1R1 knock out was about 50%. It would be good to discuss this. Which other factors could be involved in the neuroinflammation? Is it reasonable to assume that the remaining neuroinflammation does not have functional consequences?
- 9) Discussion page 17: 'it is secretion of Il1-b that propagates a positive feedback loop of consistent Il-1 activation' A feedback loop of consistent activation has not been shown here.

Referee #2 (Remarks for Author):

This is a very good and important piece of science showing that storage alone does not cause neurological problems in lysosomal storage disease, but that storage components can combine to activate neuroinflammation that does. I agree with the authors that "these data suggest that IL1 is an important mediator in the MPSIIIA inflammatory cascade, highlighting haematopoietic stem cell gene therapy using IL-1Ra as a potential anti-inflammatory therapy to treat cognitive decline in MPSIIIA and other neuronopathic lysosomal storage diseases."

However the presentation is not good and often it is hard to read. I have given some examples below. I would also commend the authors to try a schematic diagram that shows how they interpret their results and how the infammasome is activated (or not) with the different and overlapping stimuli. It would make for a very powerful paper.

Problems are:

A good list of abbreviations would make for much easier reading.

There are a lot of abbreviations some of which are not expanded in a timely fashion and many similar to each other that make it very hard to follow. This starts off with IL-1Ra in the title, continues NLRP3 in the abstract, Nlrp3, OC dot blot, IIRN, hCD11b.hIL1RN.WPRE (LV.IIL1RN, MOI, HSC, GFAP, LAMP2, DAB, DAP1, , FITC, TRITC, GAPDH, ΔΔCT GAPDH, AAV, MyD88 knockout mice in the methods, CHC, What are lin- enriched stem cells? in figs I am guessing that standard deviations, SD in the methods is the same as STDEV in the figure legends. AAV, MyD88 knockout mice, NLRP3's role in neuroinflammation should also be discussed in the introduction.

Articles missing. There are a number of these through the paper. E.g. "Lentivirus was produced and titre determined as previously described" should be, "Lentivirus was produced and the titre determined as previously described"

I do not think the phrase "according to the manufacturer's instructions" is necessary when referring to the use of kits.

Addresses required for (a gift from Dr Konstantin Dobrenis and Prof Walkley)

Results

What does "equivalent amounts based on HS quantification; Fig. EV1" mean?

"These data confirm that priming of an intracellular pro-IL-1 \bullet inflammatory response is dependent on MPSIIIA GAG acting via TLR4. This data also suggests that it is not the accumulation of MPSIIIA-specific GAG alone that is responsible for inducing the observed inflammatory response in MPSIIIA mice; instead the composition of the MPSIIIA GAG itself or a result of a co-purified GAG-binding factor must also be important." is easier to read as "These data confirm that priming of an intracellular pro-IL-1 \bullet inflammatory response is dependent on MPSIIIA GAG acting via TLR4, also suggesting that the accumulation of MPSIIIA-specific GAG alone is not responsible for inducing the observed inflammatory response in MPSIIIA mice. Instead the composition of the MPSIIIA GAG itself or a result of a co-purified GAG-binding factor must also be important." "HS which so their" is presumably "HS which shows their"

Many ideas seem to be the wrong way around, which makes the text hard to follow. E.g. I would rewrite "An ATP assay revealed significant increases in ATP levels in MPSIIIA mice compared to WT ($p < 0.05$; Fig. 3B), with ATP often released by activated immune and endothelial cells to potentiate inflammatory responses (Savage et al., 2012)" as. "ATP is often released by activated immune and endothelial cells to potentiate inflammatory responses (Savage et al., 2012). Here an ATP assay revealed significant increases in ATP levels in MPSIIIA mice compared to WT ($p < 0.05$; Fig. 3B).

"Significant increases in expression of several subunits or activators of the NLRP3 inflammasome were apparent in MPSIIIA brain tissue compared to WT (Fig. 3C; $p < 0.05$). This was confirmed with identification that caspase 1 activity was significantly increased in MPSIIIA brain homogenate (Fig. 3D; $p < 0.001$). This suggests potential involvement of the inflammasome and activation of pyroptotic cell death, which requires caspase 1 activation." Is very clumsy and hard to follow.

Discussion.

"Amyloid beta deposits have already been evidenced in MPSIIIB cortices (Ohmi et al., 2011), and have been found co-localised to HS proteoglycans in MPSIIIA post mortem brains (Ginsberg et al., 1999). Is hard to read". Try "Amyloid beta deposits have been found in MPSIIIB cortices (Ohmi et al., 2011), and also co-localised to HS proteoglycans in MPSIIIA post mortem brains (Ginsberg et al., 1999).

Untangle and rewrite the following passage "Accumulation of cholesterol occurs in various cells throughout the central nervous system and has been reported in a wide spectrum of LSDs, including not only MPS, but NiemannPick C, GM1 and GM2 gangliosidosis, and α -mannosidosis (Walkley and Vanier, 2009), similar to the cholesterol accumulation demonstrated in MPSIIIA brains here. In vitro evidence here has shown that IL-1 β secretion can be triggered via inflammasome activation with cholesterol (Rajamaki et al., 2010). In vitro studies with LPS primed monocytes and macrophages have been shown to respond to cholesterol crystals by IL-1 β secretion, in a similar fashion to our mixed glia (Rajamaki et al., 2010). Cathepsin B, caspase 1 and potassium efflux were essential for the activation of NLRP3 inflammasome by cholesterol crystals as observed here. The mechanism by which cathepsin B activates the inflammasome remains unclear, it is likely that leakage is via destabilisation of the lysosomal membrane, in response to lysosomal burden in MPSIIIA (Rajamaki et al., 2010). More recently increased cholesterol synthesis in macrophages was found to release IL-1 β via an inflammasome dependent pathway (Dang et al., 2017)."

Add to the discussion references to the data conclusions are drawn from in passages like "The central importance of an inflammatory axis initiated by the TLR4-NLRP3 pathway and IL-1 β was emphasised by our finding that in vitro stimulation of NLRP3 null mixed glia did not result in IL-1 β secretion and that glial activation was significantly reduced and behavioural abnormalities were reversed in vivo in MPSIIIA mice deficient in IL-1R1 or over-expressing LV-mediated hIL-1Ra. We observed a pronounced reduction in astrogliosis upon inhibition of IL-1 signalling; astrocytes highly express IL-1R1, allowing them to respond to changes in the levels of central IL-1 β (Srinivasan et al., 2004; Basu et al., 2004; Rothwell and Luheshi, 2000).

Referee #1 (Remarks for Author):

In the manuscript by Parker et al., the involvement of interleukin-1beta and interleukin-1 receptor antagonist in neuroinflammation induced by MPSIIIA is described. Using lentiviral gene therapy, the authors describe an approach for reduction of neuroinflammation in a mouse model. In general, the findings are interesting and provide novel insight in the pathogenesis of MPSIIIA.

Comments:

1) In the in vitro experiments using GAGs from various sources, the authors show that MPSIII GAGs specifically induced Il1b expression, whereas wt GAGs or heparin sulfate does not. While interesting, the authors do not identify the causal factor. What is in the MPSIII GAGs preparation that causes this effect?

We have added additional data which investigates the causal factor for MPSIIIA GAG immunogenicity. We have shown that it is predominantly HS as opposed to the DS/CS in the MPSIIIA GAG extract (Fig 2G), and that intracellular IL-1beta production is dependent on 2-O sulphation of MPSIIIA HS (Fig 2H and 2I). All GAG preparations were boiled to degrade protein and other cofactors. GAG preparations were also endotoxin tested via a LAL assay and showed no endotoxin contamination.

2) Fig 2B: what was the effect of GAG treatment at later time points?

We have repeated the in vivo experiments, whereby C57BL6 mice were intraperitoneally injected with PBS, WT GAG, MPSIIIA GAG, LPS, heparin or bovine kidney heparan sulphate. The levels of IL1b in the brain were measured 1 hour, 2 hours and 6 hours post injection via qPCR. Significant IL1b expression was observed 2 hours and 6 hours post-treatment (Fig 2B) when compared to a saline control. The effect peaked at 2 hours and declined by 6 hours.

3) In Fig. 2E, the authors state that no significant secretion of IL-1b into culture media was detected. However, the figure does show significant secretion.

Although significance was assigned post statistical analyses, the level of IL-1beta was below the limit of detection of the assay, and therefore should be classed as 0.

4) In Fig 3C, how was expression normalized? please show not-normalized expression levels of the genes examined as well as of the gene used for normalization

Gene expression was normalised by taking the Ct values (raw data) for the house keeping gene (GAPDH) and the gene being tested (e.g. IL-1b) and calculating the $\Delta\Delta Ct$ value (change in Ct) across these, as is standardly done in Q-PCR to standardise changes in gene expression compared to a housekeeping gene and over time (Borysiewicz et al., 2009). There is no baseline value per se as we measure the difference in expression levels against a housekeeping standard. An average Ct value was calculated for the 3 replicates. The 4 values are Gene being Tested MPS3A (T3a), Gene being Tested WT (Twt), Housekeeping Gene MPS3A (H3a), and Housekeeping Gene WT (Hwt). The differences between T3a and H3a (T3a-H3a) and Twt and Hwt (Twt-Hwt) were calculated per sample. These are our ΔCt values for the MPS3A (ΔCt_{3a}) and wildtype (ΔCt_{wt}) conditions. The difference was then calculated between ΔCt_{3a} and ΔCt_{wt} ($\Delta Ct_{3a} - \Delta Ct_{wt}$) to arrive at our delta delta Ct value ($\Delta\Delta Ct$). Since all calculations are in logarithm base 2, every time there is twice as much DNA, your Ct values decrease by 1 and will not halve. To calculate fold change the following formula was applied $2^{-\Delta\Delta Ct}$. **Normalisation of WT to 1, and MPSIIIA to WT.** An example of the raw Ct values is shown below

		Gapdh Ct	Il1b Ct
IIIa 9	99	16.51794243	33.30036926
IIIa 9	99	16.30708122	32.96333313
WT 9	223	16.17010498	35.6293602
WT 9	223	16.44099808	35.70918274
IIIa 9	104	16.57910728	32.4769516
IIIa 9	104	16.48436356	32.32744217
IIIa 9	202	15.37383556	32.94464874
IIIa 9	202	15.29645061	33.01887131
WT 9	111	15.97177505	35.08328247
WT 9	111	15.87283993	35.1925087
WT 9	133	16.53513527	35.45773315
WT 9	133	16.74427223	35.8587532
IIIa 9	112	16.65887451	33.58131409
IIIa 9	112	16.75591087	33.31336212
WT 9	206	15.7945013	35.12726593
WT 9	206	15.68466663	34.94752502
IIIa 9	133*	16.57634163	32.53634644
IIIa 9	133*	16.53100014	32.64990616
WT 9	204	14.71575069	34.39419556
WT 9	204	14.72379303	34.3925209
IIIa 9	178	16.23536873	33.63629532
IIIa 9	178	16.25427628	33.92492294
WT 9	134	16.67425346	35.67653656
WT 9	134	16.6072979	35.47218323
IIIa 9	205	17.23292351	33.97437286
IIIa 9	205	16.99202728	34.21540451
IIIa 9	221	16.46008873	33.93154144
IIIa 9	221	16.32957458	33.7898941
IIIa 9	222	15.51924515	33.7301178
IIIa 9	222	15.8452301	33.87870407
	NRT	Undetermined	Undetermined
	NRT	Undetermined	Undetermined
	NTC	Undetermined	Undetermined
	NTC	Undetermined	Undetermined

5) In Fig 4J, the VCN in brain is very low. Please comment.

There is only a low number of cells which migrate from the periphery into the brain after a bone marrow transplant. A portion of peripheral myeloid cells are able to migrate across the blood brain barrier due to the busulfan conditioning which they receive prior to the transplant (Wilkinson et al., 2013). Others from our lab have observed a therapeutic effect with regards to neuroinflammation and behaviour following HSC gene therapy with only low VCN in the brain (Sergijenko et al., 2013; Holley et al., 2018; Gleitz et al., 2018).

6) please provide a more extensive analysis of Il1b expression showing where it is expressed in the brain

We have tried to stain for IL1b in the brain, but as the cytokine is expressed at extremely low levels and works at extremely low levels it is almost impossible to do this effectively. It is also secreted, when active, which makes cellular staining less relevant. If the reviewer is referring to expression of IL-1Ra (Fig 4L), then IL-1Ra is highly secreted, so likely has effects throughout the brain, regardless of which cells produce it. Capotondo et al., 2013 have produced a very nice IHC demonstration of microglial distribution throughout the brain following transplant, therefore we would expect distribution of IL 1Ra to be also throughout the brain.

7) page 15, 3rd paragraph: the authors state that MPSIIIa x IL-1R1^{-/-} mice showed no overt differences in cognitive or neuropathological phenotype to MPSIIIa x IL-1R1^{+/+} mice. Please specify the data. It sounds strange in the light of the results of Fig. 6

This was an oversight in the text on our part. We have corrected the statement to 'WT x IL-1R1^{-/-} and MPSIIIa x IL-1R1^{-/-} mice generated were viable, however mice were housed in sterile cages due to a low level of immunodeficiency; a result of IL-1R1 knockout'.

8) Fig. 6: the reduction of astrocyte or microglial activation by overexpression of hIL-1Ra or IL-1R1 knock out was about 50%. It would be good to discuss this. Which other factors could be involved in the neuroinflammation? Is it reasonable to assume that the remaining neuroinflammation does not have functional consequences?

Systemic pentosan polysulfate treatment (a GAG mimetic and anti-inflammatory drug) and prednisolone (steroid) have both been shown to have effects on peripheral cytokine levels, as well as cognitive behaviour, indicating that peripheral inflammation may play a large role in the behavioural

abnormalities observed MPSIII murine models. We do not know whether the remaining neuroinflammation has functional consequences, the activation of microglia and astrocytes may play a protective role as opposed to destructive, we must remember that these cells will still have lysosomal burden due to GAG accumulation amongst other substrates. We have reworked the discussion to include some conjecture on this.

9) Discussion page 17: 'it is secretion of IL-1 β that propagates a positive feedback loop of consistent IL-1 activation' A feedback loop of consistent activation has not been shown here. We have reworded this, to reflect that this is our hypothesis for what is happening.

Referee #2 (Remarks for Author):

This is a very good and important piece of science showing that storage alone does not cause neurological problems in lysosomal storage disease, but that storage components can combine to activate neuroinflammation that does. I agree with the authors that "these data suggest that IL1 is an important mediator in the MPSIIIA inflammatory cascade, highlighting haematopoietic stem cell gene therapy using IL-1Ra as a potential anti-inflammatory therapy to treat cognitive decline in MPSIIIA and other neuronopathic lysosomal storage diseases."

However the presentation is not good and often it is hard to read. I have given some examples below.

We have updated the introduction and discussion to clarify some of these points and make for easier reading (*).

I would also commend the authors to try a schematic diagram that shows how they interpret their results and how the inflammasome is activated (or not) with the different and overlapping stimuli. It would make for a very powerful paper.

This was a great suggestion. A summary schematic diagram has been added to show how we think the inflammatory response is activated in MPSIIIA, and how our therapy attenuates such an inflammatory response.

1. A good list of abbreviations would make for much easier reading. There are a lot of abbreviations some of which are not expanded in a timely fashion and many similar to each other that make it very hard to follow. This starts off with IL-1Ra in the title, continues NLRP3 in the abstract, Nlrp3, OC dot blot, IIRN, hCD11b.hIL1RN.WPRE (LV.IIL1RN, MOI, HSC, GFAP, LAMP2, DAB, DAP1, , FITC, TRITC, GAPDH, $\Delta\Delta$ CT GAPDH, AAV, MyD88 knockout mice in the methods, CHC, What are lin- enriched stem cells? in figs I am guessing that standard deviations, SD in the methods is the same as STDEV in the figure legends. AAV, MyD88 knockout mice.

Abbreviations added as requested

2. NLRP3's role in neuroinflammation should also be discussed in the introduction.

This has been added

3. Articles missing. There are a number of these through the paper. E.g. "Lentivirus was produced and titre determined as previously described" should be, "Lentivirus was produced and the titre determined as previously described"

Updated as requested

4. I do not think the phrase "according to the manufacturer's instructions" is necessary when referring to the use of kits.

This has been removed

5. Addresses required for (a gift from Dr Konstantin Dobrenis and Prof Walkley)

4 and 5 – we will abide by journal requirements on these aspects

Results

6. What does "equivalent amounts based on HS quantification; Fig. EV1" mean?

We have changed the above statement to 'amounts administered based on quantity of HS detected' to clarify how much GAG was administered to mice.

7. "These data confirm that priming of an intracellular pro-IL-1• inflammatory response is dependent on MPSIIIA GAG acting via TLR4. This data also suggests that it is not the accumulation of MPSIIIA-specific GAG alone that is responsible for inducing the observed inflammatory response in MPSIIIA mice; instead the composition of the MPSIIIA GAG itself or a result of a co-purified GAG-binding factor must also be important." is easier to read as "These data confirm that priming of an intracellular pro-IL-1• inflammatory response is dependent on MPSIIIA GAG acting via TLR4, also suggesting that the accumulation of MPSIIIA-specific GAG alone is not responsible for inducing the observed inflammatory response in MPSIIIA mice. Instead the composition of the MPSIIIA GAG itself or a result of a co-purified GAG-binding factor must also be important."

We have updated the text to make this clearer

8. "HS which so their" is presumably "HS which shows their"

We have changed the above statement to 'HS which shows their'.

9. Many ideas seem to be the wrong way around, which makes the text hard to follow. E.g. I would rewrite "An ATP assay revealed significant increases in ATP levels in MPSIIIA mice compared to WT ($p < 0.05$; Fig. 3B), with ATP often released by activated immune and endothelial cells to potentiate inflammatory responses (Savage et al., 2012)" as. "ATP is often released by activated immune and endothelial cells to potentiate inflammatory responses (Savage et al., 2012). Here an ATP assay revealed significant increases in ATP levels in MPSIIIA mice compared to WT ($p < 0.05$; Fig. 3B).

See above*

10. "Significant increases in expression of several subunits or activators of the NLRP3 inflammasome were apparent in MPSIIIA brain tissue compared to WT (Fig. 3C; $p < 0.05$). This was confirmed with identification that caspase 1 activity was significantly increased in MPSIIIA brain homogenate (Fig. 3D; $p < 0.001$). This suggests potential involvement of the inflammasome and activation of pyroptotic cell death, which requires caspase 1 activation." Is very clumsy and hard to follow.

See above*

Discussion – has been rewritten with these points taken.

11. "Amyloid beta deposits have already been evidenced in MPSIIIB cortices (Ohmi et al., 2011), and have been found co-localised to HS proteoglycans in MPSIIIA post mortem brains (Ginsberg et al., 1999). Is hard to read". Try "Amyloid beta deposits have been found in MPSIIIB cortices (Ohmi et al., 2011), and also co-localised to HS proteoglycans in MPSIIIA post mortem brains (Ginsberg et al., 1999).

See above*

12. Untangle and rewrite the following passage "Accumulation of cholesterol occurs in various cells throughout the central nervous system and has been reported in a wide spectrum of LSDs, including not only MPS, but NiemannPick C, GM1 and GM2 gangliosidosis, and α -mannosidosis (Walkley and Vanier, 2009), similar to the cholesterol accumulation demonstrated in MPSIIIA brains here. In vitro evidence here has shown that IL-1 β secretion can be triggered via inflammasome activation with cholesterol (Rajamaki et al., 2010). In vitro studies with LPS primed monocytes and macrophages have been shown to respond to cholesterol crystals by IL-1 β secretion, in a similar fashion to our mixed glia (Rajamaki et al., 2010). Cathepsin B, caspase 1 and potassium efflux were essential for the

activation of NLRP3 inflammasome by cholesterol crystals as observed here. The mechanism by which cathepsin B activates the inflammasome remains unclear, it is likely that leakage is via destabilisation of the lysosomal membrane, in response to lysosomal burden in MPSIIIA (Rajamaki et al., 2010). More recently increased cholesterol synthesis in macrophages was found to release IL-1 β via an inflammasome dependent pathway (Dang et al., 2017)."

See above*

13. Add to the discussion references to the data conclusions are drawn from in passages like "The central importance of an inflammatory axis initiated by the TLR4-NLRP3 pathway and IL-1 β was emphasised by our finding that in vitro stimulation of NLRP3 null mixed glia did not result in IL-1 β secretion and that glial activation was significantly reduced and behavioural abnormalities were reversed in vivo in MPSIIIA mice deficient in IL-1R1 or over-expressing LV-mediated hIL-1Ra. We observed a pronounced reduction in astrogliosis upon inhibition of IL-1 signalling; astrocytes highly express IL-1R1, allowing them to respond to changes in the levels of central IL-1 β (Srinivasan et al., 2004; Basu et al., 2004; Rothwell and Luheshi, 2000).

See above*

2nd Editorial Decision

20 December 2019

Thank you for the submission of your revised manuscript to EMBO Molecular Medicine. We have now received the enclosed report from the referee who was asked to re-assess it. As you will see the reviewer is now supportive and I am pleased to inform you that we will be able to accept your manuscript pending minor editorial amendments including a response to the minor text change commented by referee 1.

I look forward to reading a new revised version of your manuscript as soon as possible.

***** Reviewer's comments *****

Referee #1 (Comments on Novelty/Model System for Author):

see original review

Referee #1 (Remarks for Author):

The revised manuscript has been improved significantly. The finding that it is 2-O sulphation of HS in GAGs from MPSIIIA mice is very interesting and explains the specific activity of these GAGs in activating inflammation. The authors have addressed all comments satisfactorily. Editorial suggestions:

page 15 2nd paragraph: "...suggesting that GAGs alone prime, but do not activate inflammasome responses"

this can formally not be stated at this stage of the story, I suggest to rephrase, for example replace "suggesting" by "consistent with" and refer to the experiments below that address priming.

p 15, 3rd paragraph: suggesting that GAG should read suggesting that GAGs

p16, line 1: add ref and remove comment

discussion line 1: remove comment

Figure 2H and EV1: specify 2S, 6S, and NS in legend
 Figures 2H and I are wrongly numbered in the legend

2nd Revision - authors' response

15 January 2020

Authors made the requested changes.

Corresponding Author Name: Brian Bigger

Journal Submitted to: Celine ; EMBO Molecular Medicine

Manuscript Number: EMM-2019-11185